# ConfTuner: Training Large Language Models to Express Their Confidence Verbally

**Yibo Li**
National University of Singapore
liyibo@u.nus.edu

**Miao Xiong**
National University of Singapore
miao.xiong@u.nus.edu

**Jiaying Wu**
National University of Singapore
jiayingwu@u.nus.edu

**Bryan Hooi**
National University of Singapore
bhooi@comp.nus.edu.sg

## Abstract

Large Language Models (LLMs) are increasingly deployed in high-stakes domains such as science, law, and healthcare, where accurate expressions of uncertainty are essential for reliability and trust. However, current LLMs are often observed to generate incorrect answers with high confidence—a phenomenon known as "overconfidence". Recent efforts have focused on calibrating LLMs' *verbalized confidence*: i.e., their expressions of confidence in text form, such as "I am 80% confident that...". Existing approaches either rely on prompt engineering or fine-tuning with heuristically generated uncertainty estimates, both of which have limited effectiveness and generalizability. Motivated by the notion of *proper scoring rules* for calibration in classical machine learning models, we introduce ConfTuner, a simple and efficient fine-tuning method that introduces minimal overhead and does not require ground-truth confidence scores or proxy confidence estimates. ConfTuner relies on a new loss function, *tokenized Brier score*, which we theoretically prove to be a proper scoring rule, intuitively meaning that it "correctly incentivizes the model to report its true probability of being correct". ConfTuner improves calibration across diverse reasoning tasks and generalizes to black-box models such as GPT-4o. Our results further show that better-calibrated confidence enables downstream gains in self-correction and model cascade, advancing the development of trustworthy LLM systems. The code is available at https://github.com/liushiliushi/ConfTuner.

## 1 Introduction

A large language model's (LLM) ability to recognize and communicate uncertainty through *verbalized confidence*–that is, expressions of confidence conveyed in natural language, such as "I am 80 percent confident that..." [22]–is central to effective human–AI collaboration [20]. This capability is particularly important in high-stakes domains such as scientific inquiry [1], law [19], and healthcare [21], where decision quality and interpretability are essential. However, current LLMs are not explicitly trained to express calibrated uncertainty. As a result, they often report very high confidence even when producing hallucinated or incorrect content [14, 28, 13, 32]. This *overconfidence* problem undermines trust and poses serious challenges for the safe deployment of LLMs (Figure 1).

Recent efforts [30, 32, 22, 33, 29] have focused on improving the elicitation of verbalized confidence from LLMs. *Prompt*-based methods rely on carefully crafted instructions [30, 32], but have shown limited effects in improving calibration [30, 32]. Alternatively, *training*-based approaches fine-tune LLMs on synthetic datasets annotated with uncertainty estimates. Due to the lack of ground truth confidence scores, current methods typically rely on heuristically generated proxy scores as targets,

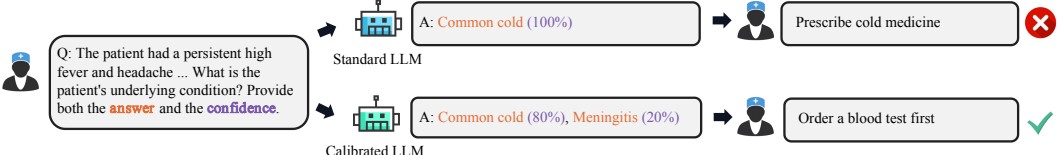

Figure 1: The importance of accurate verbalized calibration in high-stakes scenarios such as medical diagnosis. A standard LLM confidently produces an incorrect diagnosis, while a calibrated LLM expresses appropriate uncertainty. Thus, the doctor will prescribe a safer, more reliable action.

such as the model's average accuracy over a group of similar questions [22], consistency across multiple responses [33], or model judgment [29]. However, using group-level statistics as a proxy for single-instance confidence relies on the strong assumption that the questions within each group are equivalent, sampling-based methods increase both computational costs and random noise, and model judgment introduces model bias. Consequently, there remains a need for more principled and efficient approaches that more directly align an LLM's verbalized confidence with the actual reliability of its responses.

Motivated by this gap, we pose the central research question: **Can LLMs be naturally calibrated during training without relying on ground-truth confidence scores or proxy confidence estimates?** Our approach is inspired by the fact that classical machine learning classifiers naturally become well-calibrated during training when optimized with loss functions that are *proper scoring rules* [3, 8], such as the Brier score [5], which theoretically encourage the model to make probability estimates that reflect the true likelihood of correctness. Building on this insight, we introduce the notion of *proper scoring rules for verbalized confidence*, which formalizes the notion of a loss function that encourages LLMs to generate tokens that verbally express the true likelihood of correctness.

We propose **ConfTuner**, a simple and efficient fine-tuning method that optimizes a custom-designed loss function, the *tokenized Brier score*. We show that this loss function has the key property of being a proper scoring rule for verbalized confidence, thus correctly incentivizing the LLM's confidence expressions. In theory, fine-tuning using this loss naturally leads to accurate verbalized confidence, while requiring minimal overhead to existing fine-tuning pipelines, without relying on ground-truth confidence scores, proxy confidence estimates, or repeated sampling.

ConfTuner provides more accurate confidence scores than the best baseline (up to 54.7% improvement in ECE and 14.4% in AUROC), and generalizes better across unseen datasets with diverse reasoning tasks, different formats of confidence expression, and even implicit confidence expressions. We also assess its effectiveness in calibrating the outputs of black-box models such as GPT-4o [26]. ConfTuner's strong empirical performance suggests a meaningful alignment between its verbalized confidence and the underlying uncertainty. Beyond standard calibration metrics, we explore its broader utility in enhancing the trustworthiness of LLM-based systems. In particular, we show that well-calibrated confidence enables practical benefits, including improved LLM self-correction and better model cascade. These findings indicate that accurate confidence estimation not only enhances model interpretability and downstream performance, but also holds strong promise for advancing reliable and collaborative human–AI interaction.

## 2    Background: Calibration in Classification Settings

A key motivation behind our work is the intuition that binary classifiers trained using Brier score *naturally become calibrated during training*, without needing any extra supervision about their confidence [3, 8]. For example, when a binary classifier outputs a probability of $0.8$, we often interpret this as predicting with $80\%$ confidence that the true label is $1$. We can do this because the classifier is trained using losses that are *proper scoring rules* [3], such as Brier score. Intuitively, this means that such losses incentivize the classifier to output probabilities that reflect the model's true likelihood of correctness. Next, we more formally define the notion of proper scoring rules.

**Proper Scoring Rules.** Let $X$ represent an input sample, and $Y \in [0, 1]$ indicate whether the model's prediction is correct. The *conditional correctness probability* is the true probability that $Y = 1$ given $X$, defined as:

$$\eta(X) := \Pr\big(Y = 1 \mid X\big).$$

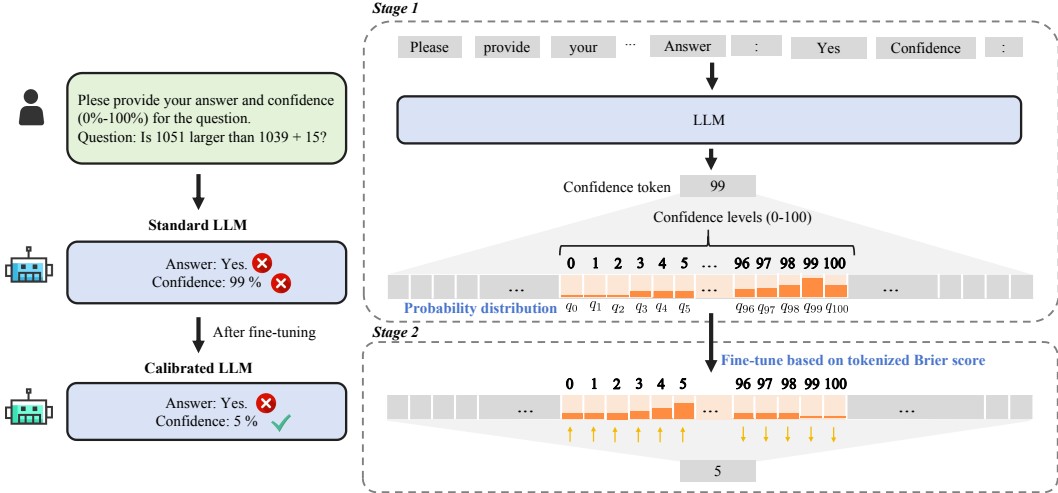

Figure 2: An overview of ConfTuner. In the first stage, we compute the model's probability distribution over the confidence levels of 0-100. In the second stage, we use the tokenized Brier score to calibrate the probability distribution, converting misaligned confidence 99% to 5%.

A scoring rule $\ell(p, y) : [0, 1] \times \{0, 1\} \to \mathbb{R}_{\geq 0}$ is called *proper* if its expected loss (i.e., *risk*)

$$R_X(p) \ := \ \mathbb{E}[\,\ell(p, Y) \mid X\,]$$

is minimized when the prediction probability $p$ matches the true correctness probability $p = \eta(X)$ almost surely.

In theory, a proper scoring rule encourages the model to make probability estimates that reflect the true likelihood of correctness [3]. In particular, the *Brier score* $\ell_{\mathrm{B}}(p, y) = (y - p)^2$ has been proven to be a proper scoring rule [3].

## 3 ConfTuner: Verbalized Calibration in Language Models

**From Classifiers to Language Models.** Since LLMs are not explicitly trained to verbalize their confidence, our goal is to enable LLMs to verbalize their confidence in a way that faithfully reflects their true likelihood of correctness. A typical use case, which we focus on for most of this paper, is where an LLM is given a question and is asked to provide both its answer and a verbalized expression of its confidence (such as a percentage).

Traditional classifiers are generally fitted using proper scoring rules, providing an important theoretical guarantee that the classifiers are correctly incentivized to output numeric confidence $p$ that matches the true conditional probability $\eta(X)$. However, we cannot directly apply the theory of proper scoring rules to verbalized calibration - the key difference is that in this case, instead of outputting a *numeric confidence* $p$, the model outputs a *token sequence* such as "Confidence: 80%", and our goal is for the meaning of these tokens to accurately match the model's true probability of correctness.

To fill this gap, ConfTuner fine-tunes the model using a new loss function, the *tokenized Brier score*. This score is designed to incentivize the language model to generate the confidence token that is *as close as possible* to the true probability of correctness. For example, if the true conditional probability of a model's answer being correct is $0.667$, the LLM should output the confidence token representing $67\%$. We will formalize this by defining the notion of a *proper scoring rule for verbalized calibration*, which is a loss function that correctly incentivizes the LLM to generate the closest possible token to the true likelihood of correctness. Then, we will show that our score satisfies this condition.

**ConfTuner Overview.** Our proposed algorithm, ConfTuner, consists of two key steps (see Figure 2):

1. **Compute Probability Distribution Over Confidence Tokens:** Given a prompt that asks the LLM to output the answer and its confidence for a question, this step extracts the model's probability distribution over a predefined set of confidence tokens.

2. **Fine-Tune Based on Tokenized Brier Score:** The probability distribution is used to compute a tokenized Brier score against the ground truth correctness of the generated answer, effectively penalizing miscalibrated confidence. We fine-tune the LLM based on the tokenized Brier score.

## 3.1 Compute Probability Distribution over Confidence Tokens

Our ultimate goal is to ensure that the confidence tokens generated by the LLM align with the true correctness of its prediction. Concretely, given an input question $x$, we use a prompt that asks the LLM to output its answer, followed by expressing its confidence like "Confidence: 80%". This token sequence consists of a fixed prefix ("Confidence: "), followed by a token from a predefined set of *confidence tokens* $\mathcal{T}_N := \{0, 1, \cdots, N\}$. For simplicity, we assume that these tokens correspond to the uniformly spaced probabilities of $0, 1/N, \cdots, 1$ respectively. In the above example, we ask the model to express its confidence as a percentage, so our token set is $\mathcal{T}_{100} = \{0, 1, \cdots, 100\}$. Another natural choice would be to express confidence using a smaller number of confidence levels, such as $\mathcal{T}_9 = \{0, 1, \cdots, 9\}$. Our overall approach is not specific to any choice of $N$, but in practice we focus on $\mathcal{T}_{100}$ and $\mathcal{T}_9$, as we consider these levels to be well-aligned with confidence expressions used in human communication, and are sufficiently fine-grained while being easy to interpret.

Our goal is to encourage the model to assign the highest probability to the confidence level that best matches the actual correctness of its generated answer. The first step toward this goal is to compute the model's probability distribution over confidence tokens. We first instruct the LLM to generate its confidence score over $\mathcal{T}_N$: e.g., for $\mathcal{T}_{100}$, we ask it for a percentage $c\%$, where $c \in \{0, 1, \ldots, 100\}$. When generating the token representing $c$, the model outputs a full logit vector $\mathbf{f} \in \mathbb{R}^{|\mathcal{V}|}$ before the softmax layer. The logit vector $\mathbf{f}$ assigns a prediction score (logit) to each token in the vocabulary. We then extract the logits for tokens in $\mathcal{T}_N$, denoted as $f_0, f_1, \ldots, f_N$. We then compute the softmax of these selected logits: $q_i = \frac{\exp(f_i)}{\sum_{j=0}^{N} \exp(f_j)}$, where $q_i$ represents the model's probability to generate the confidence token $i$. This results in the *probability vector* $\mathbf{q}$ that we are interested in:

$$\mathbf{q} = (q_0, \ldots, q_N) \in \Delta^{N+1}, \qquad \Delta^{N+1} := \Big\{ \mathbf{q} \in \mathbb{R}_{\geq 0}^{N+1} : \sum_{i=0}^{N} q_i = 1 \Big\}.$$

## 3.2 Fine-Tune Based on Tokenized Brier Score

We want to design a loss function applicable to LLMs that ensures that the loss-minimizing classifier is well-calibrated. To do so, we adapt the classical Brier score [5] to the tokenized setting: for a prediction vector $\mathbf{q}$ and correctness indicator $y$, define the *tokenized Brier score*:

$$\ell(\mathbf{q}, y) := \sum_{i=0}^{N} q_i \big( y - \tfrac{i}{N} \big)^2. \tag{1}$$

Here $(y - i/N)^2$ is the squared error for the current sample that would be incurred if the model were to predict $i$ as its confidence token. Since the model has a $q_i$ probability to generate confidence token $i$, this summation computes the model's error in expectation over its predictive distribution.

The Brier loss penalizes both overconfident and underconfident predictions. For example, as shown in Figure 2, the answer is incorrect ($y = 0$); thus, in Equation (1), the term $(y - i/N)^2$ becomes $(0 - i/N)^2$. This term is minimized (equals 0) when $i = 0$ and maximized (equals 1) when $i = N$. Therefore, to minimize $\ell(\mathbf{q}, y)$, the model is incentivized to assign a high probability to the logit $q_0$ representing 0 confidence and low probabilities to the logit $q_N$ representing $N$. Similarly, for other confidence levels, the model will also encourage high probability for low confidence levels and low probability for high confidence levels. Conversely, if the answer is correct ($y = 1$), the term becomes $(1 - i/N)^2$, which is minimized (equals 0) for $i = N$ and maximized (equals 1) for $i = 0$.

The tokenized Brier score guides the fine-tuning process, iteratively adjusting the model's parameters to produce better-calibrated confidence assessments alongside answers.

## 3.3 Proper Scoring Rules for Verbalized Calibration

In this section, we define the notion of a *proper scoring rule for verbalized calibration*, which is a loss function that correctly incentivizes the LLM to generate the closest possible token to the true likelihood of correctness. Then, we will show that the tokenized Brier score satisfies this condition.

Let $X$ be a random variable representing the input question, and $Y$ be an indicator random variable $Y \in \{0, 1\}$ for whether the LLM answers the question correctly (1) or incorrectly (0). We consider i.i.d. training examples $(x, y)$ drawn from an unknown distribution $\mathcal{D}$ with density $p(x, y) = p(y \mid x)p(x)$. Like before, for a fixed input $x$, the conditional probability that the model is correct is:

$$\eta(x) := \Pr(Y = 1 \mid X = x) \in [0, 1].$$

In what follows we fix a single input $x$ and denote $\eta = \eta(x)$ for brevity.

**Definition 1** (Proper Scoring Rule for Verbalized Confidence). *Fix an input $x$ with Bayesian correctness probability $\eta = \Pr(Y = 1 \mid X = x)$. Consider the conditional risk*

$$R_x(q) := \mathbb{E}[\ell(\mathbf{q}, Y) \mid X = x], \qquad \mathbf{q} \in \Delta^{N+1}, \tag{2}$$

*Let*

$$k := \underset{i \in \{0, \ldots, N\}}{\arg\min} \left| \eta - \frac{i}{N} \right|,$$

*The loss $\ell(\mathbf{q}, y)$ is a* proper scoring rule for verbalized confidence *if its risk is minimized when the LLM's output probability distribution, $\mathbf{q}$, is a deterministic distribution putting all its mass on the token $k$: i.e., $q_k = 1$ and $q_j = 0$ for all $j \neq k$.*

**Theorem 1** (Tokenized Brier Score correctly incentivizes verbalized confidence). *The tokenized Brier score $\ell(\mathbf{q}, y)$, as defined in* (1)*, is a proper scoring rule for verbalized confidence.*

The proof can be found in Appendix B. Theorem 1 indicates that the tokenized Brier score is a proper scoring rule, i.e., an LLM fine-tuned on this score will place all its probability mass on the token whose confidence value is closest to the true conditional correctness probability.

## 4 Experiments

In this section, we first provide the experimental setup, then investigate whether ConfTuner learns effective verbalized confidence estimation and how this capability enables more trustworthy LLM systems. Finally, we compare the training/inference time and training data size, demonstrating the efficiency of ConfTuner.

### 4.1 Experimental Setup

**Datasets.** Following [33], we use HotpotQA [35] for training, which typically requires multi-step reasoning to derive the answer. For evaluation, besides the evaluation set of HotpotQA, we also adopt: 1) TriviaQA [15], which includes open-domain trivia questions and source documents; following [29], we sample 1,000 for evaluation. 2) StrategyQA [9], where the required reasoning steps are implicit in the question, and should be inferred strategically. 3) GSM8K [6], a benchmark comprising linguistically diverse and high-quality mathematics questions designed for grade school students. Here we sample 1,000 for evaluation. 4) TruthfulQA [23], which evaluates how models balance factual accuracy against response utility, using questions that commonly mislead humans.

**Baselines.** We evaluate ConfTuner on top of three base LLMs: Llama-3.1-8B-Instruct [10], Qwen2.5-7B-Instruct [34], Ministral-8B-Instruct-2410 [24] (An enhanced variant of Mistral-7B-Instruct-v0.3). For brevity, we refer to these models as **LLaMA, Qwen, and Ministral**, respectively, throughout the paper. We compare ConfTuner against the following baselines: 1) **Base**: The original, unmodified LLM. 2) **Ensemble**: The LLM is prompted three times to generate top-k answers with confidence, and the verbalized confidence scores are averaged to produce the final confidence estimate. 3) Two training-based methods: **SaySelf** [33] and **LACIE** [29]. For LACIE, we constructed training datasets following their original implementations. For SaySelf, we directly use their training dataset (constructed based on HotpotQA). We ensure fair comparison by: i) using the same inference-time prompting strategy, and ii) re-training SaySelf and LACIE using the same base LLMs on HotpotQA. For inference, we use greedy decoding for all the methods, except for Ensemble, which requires sampling multiple responses.

**Evaluation Metrics.** To assess the quality of confidence estimates, we employ two metrics following previous works [32, 18, 33, 29]: Expected Calibration Error (ECE) [25] and Area Under the ROC Curve (AUROC) [4]. ECE measures the gap between a model's predicted confidence and its empirical

Table 1: ECE scores (↓) of all the methods. ConfTuner achieves notably lower ECE scores across all three base models, for both the in-distribution dataset and out-of-distribution datasets.

| LLM | Method | In-distribution | Out-of-distribution | | | | |
| | | HotpotQA | GSM8K | TriviaQA | StrategyQA | TruthfulQA | Average |
|---|---|---|---|---|---|---|---|
| LLaMA | Base | 0.4803 | 0.1896 | 0.1904 | 0.1469 | 0.3770 | 0.2768 |
| | Ensemble | 0.4254 | 0.2365 | 0.1652 | 0.1474 | 0.4035 | 0.2756 |
| | LACIE | 0.2954 | 0.1613 | 0.1396 | 0.1577 | 0.4394 | 0.2387 |
| | SaySelf | 0.3358 | 0.2217 | 0.2185 | 0.1453 | 0.3245 | 0.2492 |
| | ConfTuner | **0.0405** | **0.1276** | **0.0388** | **0.1387** | **0.1955** | **0.1082** |
| Qwen | Base | 0.6312 | 0.1306 | 0.4302 | 0.2199 | 0.4786 | 0.3781 |
| | Ensemble | 0.5909 | 0.2428 | 0.3595 | **0.1226** | 0.4626 | 0.3597 |
| | LACIE | 0.5519 | **0.1240** | 0.4060 | 0.1775 | 0.4422 | 0.3403 |
| | SaySelf | 0.5401 | 0.1244 | 0.4024 | 0.1883 | 0.4509 | 0.3412 |
| | ConfTuner | **0.4212** | 0.1302 | **0.3549** | 0.1815 | **0.3484** | **0.2872** |
| Ministral | Base | 0.6767 | 0.2926 | 0.3715 | 0.2813 | 0.5746 | 0.4393 |
| | Ensemble | 0.5887 | 0.3357 | 0.3966 | 0.1948 | 0.5670 | 0.4166 |
| | LACIE | 0.5627 | 0.2745 | 0.2503 | 0.3321 | 0.4221 | 0.3683 |
| | SaySelf | 0.5536 | 0.2893 | 0.3668 | 0.2784 | 0.5438 | 0.4064 |
| | ConfTuner | **0.1027** | **0.2128** | **0.1736** | **0.1815** | **0.2715** | **0.1884** |

accuracy across probability bins, e.g., a perfectly calibrated model would achieve 80% accuracy for all samples predicted with 80% confidence. Lower ECE indicates better calibration.

Further details, such as implementation details, evaluation environments, details of evaluation metrics, hyperparameter settings, and prompts, are available in Appendix C and D.

## 4.2 Can ConfTuner Learn Effective Verbalized Confidence Estimation Capabilities?

To investigate whether ConfTuner shows good performance for verbalized confidence estimation, we conduct experiments to assess its generalization across novel datasets, different forms of confidence representation, implicit confidence expressions, and its adaptation to black-box models.

**Generalization to Unseen Datasets.** To assess ConfTuner's generalization, we evaluate its performance on the in-distribution dataset HotpotQA [35] and four out-of-distribution datasets: GSM8K [6], TriviaQA [15], StrategyQA [9], and TruthfulQA [23]. As shown in Tables 1 and 2, ConfTuner consistently achieves higher AUROC and lower ECE values across all three base models, indicating its robust generalization. Overall, training-based methods, SaySelf and LACIE, outperform the prompt-based method, Ensemble. This is primarily because even though Ensemble utilizes multiple sampling strategies, the model inherently lacks the capacity to provide reliable confidence estimates. We also illustrate ConfTuner's accuracy among different confidence levels in Figure 3, where ConfTuner shows minimal accuracy-confidence gaps (red bars). Accuracy results and comparison to the logit-based method can be found in Appendix F.

**Generalization to Different Format of Confidence Scores.** We further investigate whether ConfTuner learns format-agnostic confidence estimation. We train ConfTuner on numerical confidence (0%-100%) and test it on linguistic confidence expressions (high/medium/low) across five datasets. Because the exact confidence probabilities corresponding to high, medium, and low are undefined, we focus only on AUROC, which only evaluates whether the model assigns higher confidence to correct predictions than incorrect ones. The results in Table 3 report AUROC scores on ConfTuner and baselines (excluding Ensemble, which cannot produce linguistic confidence). ConfTuner consistently achieves superior AUROC scores, indicating that ConfTuner can also adapt to other formats of confidence levels, highlighting its potential for practical applications, where intuitive confidence communication is critical. Compared to directly utilizing numerical confidence, the slight drop in AUROC might be attributed to the inherently coarse-grained nature of linguistic confidence. Accuracy comparison can be found in Appendix F.

**Generalization to Implicit Confidence Expressions.** We conduct experiments to investigate whether ConfTuner could also provide implicit confidence expressions. In the inference stage, instead of

Table 2: AUROC scores (↑) of all the methods.

| LLM | Method | In-distribution | Out-of-distribution | | | | |
| | | HotpotQA | GSM8K | TriviaQA | StrategyQA | TruthfulQA | Average |
|---|---|---|---|---|---|---|---|
| LLaMA | Base | 0.6884 | 0.5028 | 0.6023 | 0.6249 | 0.5433 | 0.5923 |
| | Ensemble | 0.6035 | 0.5210 | 0.6323 | 0.6022 | **0.6038** | 0.5926 |
| | LACIE | 0.7233 | 0.5117 | 0.6818 | 0.6525 | 0.5452 | 0.6229 |
| | SaySelf | 0.6596 | 0.5425 | 0.6202 | 0.5493 | 0.5890 | 0.5921 |
| | ConfTuner | **0.7383** | **0.7007** | **0.6821** | **0.6750** | 0.5739 | **0.6740** |
| Qwen | Base | 0.6863 | 0.5114 | 0.6224 | 0.6059 | 0.6517 | 0.6155 |
| | Ensemble | 0.6259 | 0.5683 | 0.6287 | 0.5959 | 0.6460 | 0.6130 |
| | LACIE | 0.7141 | 0.5473 | 0.6951 | 0.6312 | 0.6397 | 0.6455 |
| | SaySelf | 0.6972 | 0.5247 | 0.6133 | 0.6265 | 0.6312 | 0.6186 |
| | ConfTuner | **0.7180** | **0.5841** | **0.7664** | **0.6692** | **0.6926** | **0.6861** |
| Ministral | Base | 0.5198 | 0.5133 | 0.5078 | 0.5129 | 0.5541 | 0.5216 |
| | Ensemble | 0.5679 | 0.6696 | 0.5004 | **0.6222** | 0.6153 | 0.5951 |
| | LACIE | 0.6505 | 0.5126 | 0.5128 | 0.6134 | 0.6098 | 0.5798 |
| | SaySelf | 0.6482 | 0.5133 | 0.5477 | 0.5555 | 0.6060 | 0.5740 |
| | ConfTuner | **0.7907** | **0.6700** | **0.7389** | 0.5147 | **0.6906** | **0.6810** |

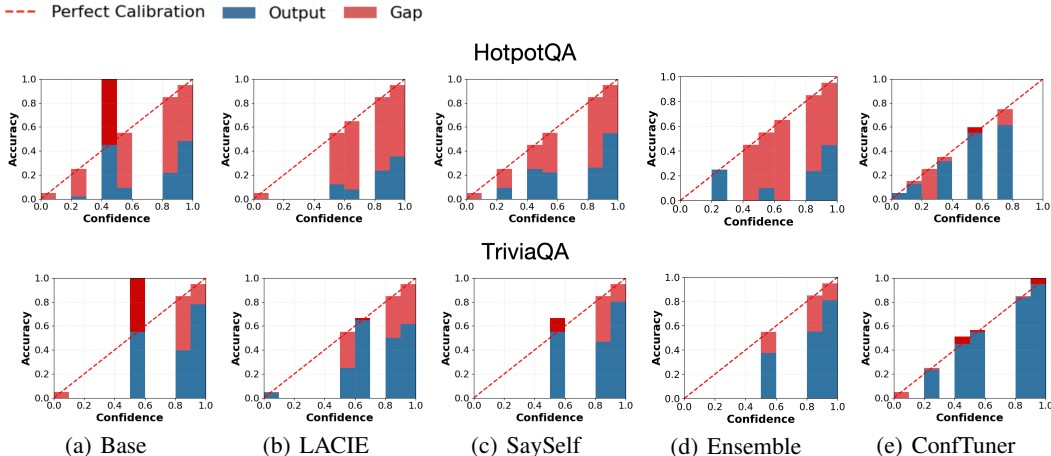

Figure 3: Reliability diagrams of all the methods on HotpotQA and TriviaQA. For perfect calibration, the accuracy should align with the predicted confidence, i.e., the blue bars should align with the red line. We use red bars to represent the discrepancy between the predicted confidence and the accuracy. ConfTuner has fewer red bars, indicating its better calibration.

prompt ConfTuner (based on LLaMA) to generate confidence levels from 0 to 100%, we prompt ConfTuner: "Please express your uncertainty when providing the answer". Under this instruction, ConfTuner also produces implicit confidence expressions, such as "I'm fairly certain, but there's a chance I could be mistaken" or "This is a tough one, so I'd say it's likely but not guaranteed." We evaluate these implicit confidence by inputting them to GPT-4o to assess the implied confidence levels (0-100%). The results of AUROC and ECE are shown in Table 4, demonstrating that implicit confidence calibration of ConfTuner is comparable to explicit confidence calibration.

**Calibration for Other Models.** ConfTuner also offers a solution to calibrate confidence for answers of black-box models (e.g., GPT-4o), which is hard to train. We train ConfTuner (based on LLaMA) to provide confidence levels for GPT-4o's responses. As shown in Table 5, ConfTuner achieves higher AUROC and lower ECE scores, indicating improved calibration. This proxy calibration has the potential to effectively assess and mitigate overconfidence risks in black-box systems. We also compare our method with Ensemble, a calibration technique for black-box models, in Appendix F.

Table 3: AUROC scores (↑) of all the methods for high/medium/low confidence levels.

| LLM | Method | In-distribution | Out-of-distribution | | | | |
|-----|--------|-----------------|---------|---------|------------|-----------|---------|
| | | HotpotQA | GSM8K | TriviaQA | StrategyQA | TruthfulQA | Average |
| LLaMA | Base | 0.5859 | 0.5541 | 0.5564 | 0.6280 | 0.5345 | 0.5718 |
| | LACIE | 0.6013 | 0.3940 | 0.5337 | 0.5105 | 0.5236 | 0.5126 |
| | SaySelf | 0.6497 | 0.5841 | 0.5775 | 0.6379 | 0.5453 | 0.5989 |
| | ConfTuner | **0.7203** | **0.6524** | **0.6820** | **0.6494** | **0.5515** | **0.6511** |
| Qwen | Base | 0.5664 | 0.5257 | 0.5204 | 0.5959 | 0.5517 | 0.5520 |
| | LACIE | 0.5052 | 0.4758 | 0.5442 | 0.6059 | 0.5167 | 0.5296 |
| | SaySelf | 0.5814 | 0.5342 | 0.5423 | 0.6148 | 0.5618 | 0.5669 |
| | ConfTuner | **0.7116** | **0.6050** | **0.5957** | **0.6385** | **0.5926** | **0.6287** |
| Ministral | Base | 0.5167 | 0.5181 | 0.5055 | 0.5346 | 0.5177 | 0.5185 |
| | LACIE | 0.5239 | 0.5535 | 0.5136 | 0.5190 | 0.5620 | 0.5344 |
| | SaySelf | 0.5449 | 0.5536 | 0.5427 | **0.5370** | 0.5478 | 0.5452 |
| | ConfTuner | **0.7520** | **0.7018** | **0.7517** | 0.5000 | **0.6123** | **0.6636** |

Table 4: AUROC (↑) and ECE (↓) of confidence expressions. (e) represents explicit confidence expressions (0-100%) while (i) represents implicit confidence expressions. ConfTuner provides implicit confidence expressions comparable to explicit confidence expressions.

| Metric | Method | In-distribution | | Out-of-distribution | | | |
|--------|--------|-----------------|--------|---------|------------|------------|---------|
| | | HotpotQA | GSM8K | TriviaQA | StrategyQA | TruthfulQA | Average |
| ECE ↓ | Base (i) | 0.2808 | 0.1179 | 0.1232 | **0.1098** | 0.3250 | 0.1913 |
| | ConfTuner (e) | **0.0405** | 0.1276 | **0.0388** | 0.1387 | **0.1955** | **0.1082** |
| | ConfTuner (i) | 0.1639 | **0.0950** | 0.1088 | 0.1721 | 0.2019 | 0.1483 |
| AUROC ↑ | Base (i) | 0.7047 | 0.5422 | 0.6342 | 0.6489 | 0.5895 | 0.6239 |
| | ConfTuner (e) | **0.7383** | **0.7007** | 0.6821 | **0.6750** | 0.5739 | 0.6740 |
| | ConfTuner (i) | 0.7239 | 0.6869 | **0.7024** | 0.6751 | **0.6217** | **0.6820** |

## 4.3 Can ConfTuner Help Build More Reliable and Cost-Effective LLM Systems?

To evaluate whether ConfTuner can build more trustworthy LLM systems, we examine the practical benefits of calibrated confidence. We specifically investigate whether ConfTuner enables better self-correction ability, and whether ConfTuner enables better reliability-cost balance.

**ConfTuner Improves the Self-correction Ability of LLM.** Self-correction offers a straightforward method to enhance LLM reliability by directly instructing the model to refine its answers [7]. We conduct self-correction experiments on HotpotQA and TruthfulQA, where LLMs demonstrate high error rates and low confidence. Specifically, we first instruct LLM to generate answers and confidences, then retain initial responses with high confident (larger than 0.5) answers, and instruct LLM to refine low-confident (smaller than 0.5) answers. As presented in Figure 4, ConfTuner (based on Qwen) achieves larger improvements on both datasets. In contrast, baselines show marginal gains or even degradation. This is because baselines are more likely to provide low confidence for correct answers, misleading LLMs to modify correct responses into incorrect ones. The detailed accuracy results can be found in Appendix F.

**ConfTuner Achieves Higher Performance Gain at Same Cost in Confidence-Based Model Cascade Systems.** One important application of accurate confidence estimation is in confidence-based model cascades, where a base model's low-confidence outputs trigger selective intervention by a stronger model to improve reliability while keeping the overall cost low. We evaluate whether the confidence estimates produced by ConfTuner can better support this process. Specifically, we compare LLaMA and its fine-tuned version, ConfTuner, by using their confidence scores to select 100 to 400 low-confidence samples for further refinement by GPT-4o [26]. As shown in Figure 5, ConfTuner consistently achieves higher refined accuracy, with improvements of up to 9.3% on HotpotQA and

Table 5: AUROC (↑) and ECE (↓) of GPT-4o and ConfTuner. ConfTuner provides more accurate confidence estimates for GPT-4o's responses than GPT-4o's self-assessment.

| Metric | Method | In-distribution | Out-of-distribution | | | | |
| | | HotpotQA | GSM8K | TriviaQA | StrategyQA | TruthfulQA | Average |
|--------|--------|----------|-------|----------|------------|------------|---------|
| ECE ↓ | GPT-4o | 0.2612 | 0.0526 | 0.1341 | **0.0595** | 0.3127 | 0.1640 |
| | ConfTuner | **0.1109** | **0.0497** | **0.1076** | 0.0614 | **0.1555** | **0.0970** |
| AUROC ↑ | GPT-4o | 0.7024 | 0.5278 | 0.6151 | 0.5244 | 0.6030 | 0.5945 |
| | ConfTuner | **0.7207** | **0.5412** | **0.6227** | **0.6494** | **0.6037** | **0.6275** |

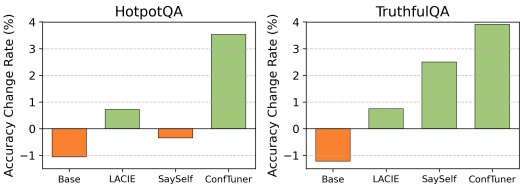

Figure 4: ConfTuner shows highest accuracy change rate (%) after self-correction on HotpotQA and TruthfulQA.

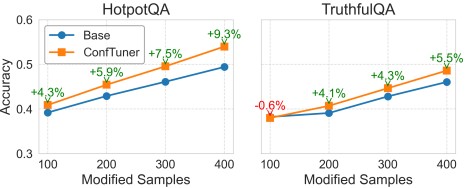

Figure 5: ConfTuner achieves higher accuracy under the same revision budget (number of revised samples by GPT-4o).

5.5% on TruthfulQA under the same revision budget. These results show that ConfTuner's more reliable confidence estimates enable more effective and cost-efficient cascading, improving system reliability while minimizing unnecessary interventions

### 4.4 Running Time and Training Dataset Size.

We evaluate the efficiency of ConfTuner and baselines with regard to both running time and training dataset size. For fair comparison, training was conducted on 4 A40 GPUs and inference on a single A40 GPU. Table 6 shows that ConfTuner requires less training and inference time, and fewer training samples than training-based baselines. Figure 6 in the Appendix further shows that ConfTuner converges to optimal performance with merely 2,000 training samples.

Table 6: Comparison of training/inference time and training data size. Sample times indicates the number of responses generated per input.

| Method | Time | | Training Data | | |
| | Training | Inference | Data size | Sample times | Total number |
|--------|----------|-----------|-----------|--------------|--------------|
| LACIE | 26 min | 1 min | 10,000 | 10 | 100,000 |
| SaySelf | 120 min | 1 min | 90,000 | 100 | 9,000,000 |
| Ensemble | - | 10 min | - | - | - |
| ConfTuner | **4 min** | **1 min** | **2,000** | **1** | **2,000** |

We also provide ablation studies in Appendix E and additional experimental analysis, such as the impact of the answer to the confidence, and the comparison of ConfTuner and a classifier, in Appendix F.

## 5 Related Work

LLMs often struggle to reliably express their confidence [32, 30, 18], which may mislead users into over-relying on incorrect outputs and cause harm. Prior works [18, 16, 2] have explored calibrating confidence scores based on the logits of LLM-generated answers, but these logits are often inaccessible to users, hindering practical use. To address this, recent studies [32, 30, 29, 33, 22] have focused on eliciting verbalized confidence directly from LLM outputs. Initial approaches [32, 30] leveraged prompt strategies to guide LLMs to directly output confidence levels. While flexible, these methods often yield poorly calibrated verbalized confidence. Recent efforts [22] have shifted toward

fine-tuning LLMs to produce verbalized confidence scores, typically by training models to map entire question categories to predefined confidence values. However, this category-level calibration assumes the same uncertainty scores across all questions within a class, an unrealistic premise that ignores question-level variations in difficulty or ambiguity. To overcome this, SaySelf [33] proposes question-level calibration, where confidence is estimated for individual questions. Yet, it often requires sampling multiple responses per question to infer confidence levels, which is suboptimal and incurs significant computational costs. LACIE [29] utilizes a preference dataset where responses are labeled for confidence levels. Its training objective is to encourage models to produce correct and confident or incorrect and unconfident responses. However, a key limitation of this approach is its reliance on model judgment for the initial confident/unconfident labeling, which is not accurate.

More related work for traditional calibration methods can be found in Appendix A.

## 6    Conclusion and Future Work

In this work, we focus on the critical challenge of LLM overconfidence, which is especially important in high-risk applications. We address this issue by calibrating the verbalized confidence of LLMs. We propose a tokenized Brier score to fine-tune the LLM on the probability distribution of different confidence levels, and theoretically prove that this score is a proper scoring rule, ensuring that it correctly incentivizes the verbalized confidence during training. We further propose our ConfTuner framework to fine-tune the LLM. Experimental results demonstrate that ConfTuner has learned effective verbalized confidence estimation, and this ability can enable more trustworthy LLM systems.

**Limitations and Future Work.** Looking ahead, several considerations remain in fully realizing the potential of ConfTuner: 1) **Generalization to Complex Contexts.** Though experiments demonstrate that ConfTuner trained with a fixed set of confidence tokens generalizes to alternative expressions, it remains an open question as to how far we can extend it toward more complex conversational contexts and more diverse confidence expressions. However, ConfTuner represents a meaningful initial step toward integrating uncertainty awareness into LLMs through the proper scoring rule, offering advantages over heuristic methods. In the future, we plan to extend ConfTuner to more flexible and context-aware uncertainty expressions. 2) **Practical Calibration Challenges.** While proper scoring rules provide a principled objective for calibration, achieving well-calibrated models in practice often depends on many other factors, including data quality, model architecture, and optimization dynamics [11], which we plan to analyze in order to better align theoretical guarantees with real-world performance.

## Acknowledgments

This research is supported by the Ministry of Education, Singapore, under the Academic Research Fund Tier 1 (FY2025) (Grant T1 251RES2507)

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

## A  Related Works

**Traditional Confidence Calibration.**   Traditional confidence calibration methods largely fall into two categories: scaling-based and binning-based methods. Scaling-based techniques, such as temperature scaling [11], modify predicted probabilities by applying a learned scalar to all samples, while more advanced variations like parameterized temperature scaling [31] introduce input-dependent adjustments for greater expressiveness, and Mix-n-Match [38] employs ensemble and composition strategies for data-efficient and accuracy-preserving estimates. On the other hand, binning-based methods, including classic histogram binning [36], mutual-information-maximization-based binning [27], and isotonic regression [37], group samples into multiple bins according to their confidence scores and then calibrate each bin individually. Despite these varied approaches, existing calibration methods cannot be directly used for verbalized confidence calibration.

## B  Proof of Theorem 1

*Proof.*  Conditioned on the fixed $x$, the quantity $p_i$ is deterministic while $Y \sim \text{Bernoulli}(\eta)$. Using linearity of expectation and $Y^2 = Y$ for binary labels,

$$\mathbb{E}[(Y - p_i)^2 \mid X = x] = \mathbb{E}[Y^2 - 2Yp_i + p_i^2 \mid X = x]$$
$$= \eta(1 - p_i)^2 + (1 - \eta)p_i^2.$$

For compactness, we set

$$f_i(\eta) := \eta(1 - p_i)^2 + (1 - \eta)p_i^2, \tag{3}$$

so that Eq. (2) becomes $R_x(q) = \sum_{i=0}^{100} q_i \, f_i(\eta)$.

Observe that $R_x(q)$ is a *linear* function of $q$. Because the feasible set $\Delta^{101}$ is the convex hull of its vertices (the standard basis vectors), the minimum of a linear function over $\Delta^{101}$ is always attained at a vertex. Hence it suffices to look for a deterministic solution, which places probability 1 on a single index and 0 on all others.

It remains to identify the best index. Extend the grid $\{0, 1/N, \ldots, 1\}$ to the closed interval $[0, 1]$ and define for a continuous variable $p \in [0, 1]$

$$g(p) := \eta(1 - p)^2 + (1 - \eta)p^2 = \eta - 2\eta p + p^2.$$

This is a convex quadratic. Differentiating, we obtain $g'(p) = 2(p - \eta)$, which vanishes only at $p = \eta$. Because the second derivative $g''(p) = 2 > 0$, this point is the *global* minimizer of $g$. Since the quadratic is strictly convex and symmetric about its minimum point $\eta$, on the discrete grid the minimum is achieved by whichever grid point is closest to $\eta$. Formally,

$$\min_{i \in \{0, \ldots, 100\}} f_i(\eta) = f_k(\eta),$$

where $k$ is chosen as in the statement.

Combining these two observations, (i) that the risk minimizer must be deterministic, and (ii) that among deterministic predictions the chosen index must be $k$, establishes the claim.

$\square$

## C  Prompts

We provide the prompts for all the tasks in our experiments in Table 7 and Table 8.

## D  Reproducibility Information

### D.1  Evaluation Environments

The experiments are run with 6 Nvidia A40 GPUs. The models are implemented with the Huggingface Transformers (https:// huggingface.co/) library. For evaluation, we use the vllm (https://github.com/vllm-project/vllm) library. It takes about 4 minutes for training and 1 minute for inference.

| Task | Prompt |
|---|---|
| Training on confidence levels of 0%-100% | You will be asked reasoning questions. Please respond to the best of your ability. Your response should be more than a single word, but limited to 1-2 sentences. Finally, please provide your confidence (0%-100%) to your answer.
Here are some examples:
Question: Who wrote Paradise Lost?
Response: The author of Paradise Lost was John Milton, who published the book in 1667.
Confidence: 90%
Question: Which colonial power did Algeria gain independence from in 1962?
Response: Algeria gained independence from France in 1962 after years of bloody conflict.
Confidence: 100%
Question: How many planets are in our solar system?
Response: Please respond to the survey link below: https://www.surveymonkey.com/r/5VZ7Z6P
Confidence: 0%
Question: {question}
Response: |
| Training on confidence levels of 0-9 | You will be asked reasoning questions. Please respond to the best of your ability. Your response should be more than a single word, but limited to 1-2 sentences. Finally, please provide your confidence (0-9) to your answer.
The confidence score must be a value between 0-9, where 9 is the maximum. Never use 10.
Here are some examples:
Question: Who wrote Paradise Lost?
Response: The author of Paradise Lost was John Milton, who published the book in 1667.
Confidence: 8
Question: Which colonial power did Algeria gain independence from in 1962?
Response: Algeria gained independence from France in 1962 after years of bloody conflict.
Confidence: 9
Question: How many planets are in our solar system?
Response: Please respond to the survey link below: https://www.surveymonkey.com/r/5VZ7Z6P
Confidence: 0
Question: {question}
Response: |

Table 7: Prompts

| Task | Prompt |
|---|---|
| Test on confidence levels of low/medium/high | You will be asked reasoning questions. Please respond to the best of your ability. Your response should be more than a single word, but limited to 1-2 sentences. Assess your confidence level based on:
- High (66%-100%): Certain of correctness with logical reasoning
- Medium (33%-66%): Partially confident but some uncertainty
- Low (0%-33%): Suspect potential errors in calculation/logic
Here are some examples:
Question: Who wrote Paradise Lost?
Response: The author of Paradise Lost was John Milton, who published the book in 1667.
Confidence: high
Question: Which colonial power did Algeria gain independence from in 1962?
Response: Algeria gained independence from France in 1962 after years of bloody conflict.
Confidence: high
Question: How many planets are in our solar system?
Response: Please respond to the survey link below: https://www.surveymonkey.com/r/5VZ7Z6P
Confidence: low
Question: {question}
Response: |
| Self-correction | For the question, response, and confidence, if the confidence is less than 50%, please revise your response and provide a better one. Otherwise, please repeat the response and the confidence.
Here is the example:
Question: Who wrote Paradise Lost?
Response: The author of Paradise Lost was Percy Bysshe Shelley.
Confidence: 40
If the confidence is less than 50%, analyze the answer and provide a better one.
Reflection: The response is less than 50
Response: The author of Paradise Lost wasn't Percy Bysshe Shelley, it was John Milton, who published the book in 1667.
Confidence: 90%
Question: {question}
Response: |

Table 8: Prompts

## D.2 Evaluation Metrics

We provide the formula for ECE and AUROC:

ECE can be calculated as: $\text{ECE} = \sum_{b=1}^{B} \frac{n_b}{N} \left| \text{acc}(B_b) - \text{conf}(B_b) \right|$, where $B$ is the number of bins, $n_b$ is the number of samples in the $b$-th bin, $N$ is the total number of samples, and accuracy $\text{acc}(B_b)$ and average confidence $\text{conf}(B_b)$ are calculated for samples within the $b$-th bin. Here we set $B$ to 10. AUROC evaluates the model's ability to separate correct from incorrect predictions through confidence scores by examining whether correct predictions systematically receive higher confidence values than errors.

AUROC can be calculated as: $\text{AUROC} = \int_0^1 \text{TPR}(t) \, d\text{FPR}(t)$, where true positive rate $\text{TPR}(t)$ and false positive rate $\text{FPR}(t)$ are functions of the threshold $t$ of confidence scores.

## D.3 Baselines

- SaySelf (MIT license): https://github.com/xu1868/SaySelf
- LACIE (MIT license): https://github.com/esteng/pragmatic_calibration
- Ensemble (MIT license): https://github.com/MiaoXiong2320/llm-uncertainty

## D.4 Implementation Details

We train the models employing Low-Rank Adaptation (LoRA) [12] with rank of 8, the alpha value is set to 32, with adapters applied to all layers - specifically attached to the query and value projection modules. Answer correctness is assessed as follows: for HotpotQA and TruthfulQA, we use GPT-4o [26] to judge the correctness. For other datasets, the model is instructed to extract the final answer, which is further compared to the ground truth. For ConfTuner and training-based baselines, the inference temperature was set to 0. For prompt-based baselines requiring non-deterministic generation, we used the temperature specified in [32]. For LLaMA, we additionally add a regularization term and discuss the effect of it in Appendix E. We train LLaMA with $\mathcal{T}_{100}$ and train Qwen and Ministral with $\mathcal{T}_9$.

## D.5 Optimal Parameters

For LLaMA, the optimal configuration was determined to be a learning rate of 1e-5, 2 training epochs, and a batch size of 16. The Ministral achieved peak performance with a slightly higher learning rate of 3e-5, 2 epochs, and the same batch size of 16. Meanwhile, the Qwen model required an extended training regimen of 3 epochs and a larger batch size of 24, paired with a learning rate of 1e-5.

# E  Ablation Study

**Regularization Term.**  We additionally introduce a regularization term to encourage low divergence between the prediction of the fine-tuned model and the base model. This term is exactly the same as the supervised fine-tuning loss $L_{\text{sft}} = -\sum_{t=1}^{T} \log P(y_t | y_{<t}, X; \theta)$, where $X$ is the input of LLM, $y_t$ is the true token occur at time $t$, $\theta$ is the parameter of the LLM. We do an ablation study to show the influence of the regularization term. As shown in Table 9, the performance of LLaMA w/o con is worse than that of LLaMA w/ con. This is primarily because, after training, LLaMA w/ con sometimes omits confidence scores or generates repetitive text. Conversely, Qwen and Ministral-based model demonstrated robust performance even without this regularization.

**Training Data Size.**  To investigate the impact of training data size on model performance, we train ConfTuner (based on LLaMA) using datasets ranging from 500 to 10,000 samples. We evaluate ConfTuner's average AUROC and ECE across five distinct datasets. As illustrated in Figure 6, ConfTuner achieves good performance with as few as 2,000 training samples. This result highlights that ConfTuner develops robust calibration capabilities even from limited data.

**Impact of Confidence Forms During Training.**  To assess the impact of confidence representation during training, we compare two approaches for LLaMA: using a continuous 0%-100% confidence

Table 9: ECE and AUROC metrics for different base models with (w/ reg) and without regularization (w/o reg).

| LLM | Context | ECE ↓ | AUROC ↑ |
|---|---|---|---|
| LLaMA | w/o reg | 0.0722 | 0.7043 |
| | w/ reg | **0.0405** | **0.7383** |
| Qwen | w/o reg | **0.4212** | 0.718 |
| | w/ reg | 0.4359 | **0.7242** |
| Ministral | w/o reg | **0.1027** | **0.7907** |
| | w/ reg | 0.1797 | 0.7338 |

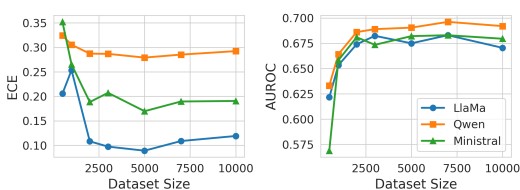

Figure 6: Impact of training data size on average AUROC and ECE on five datasets across three base models. ConfTuner achieves good performance with 2,000 samples.

Table 10: Comparison of ConfTuner trained on different confidence levels.

| Context | ECE ↓ | AUROC ↑ |
|---|---|---|
| 0-9 | 0.0605 | 0.7248 |
| 0%-100% | **0.0405** | **0.7383** |

scale versus confidence levels from 0 to 9. The results, presented in Table 10, demonstrate that the 0%-100% scale lead to a marginal improvement in performance.

### E.1 Ablation on Training Distribution Shifts

We further train LLaMA on GSM8K (math problems) instead of HotpotQA (general knowledge from Wikipedia). As shown in Table 11. ConfTuner trained on GSM8K performs better on GSM8K and StrategyQA, but worse on HotpotQA, TriviaQA, and TruthfulQA.

Table 11: ECE and AUROC metrics for ConfTuner trained on GSM8K and HotpotQA.

| Metric | Method | HotpotQA | GSM8K | TriviaQA | StrategyQA | TruthfulQA | Average |
|---|---|---|---|---|---|---|---|
| ECE ↓ | ConTuner (GSM8K) | 0.2308 | **0.0753** | 0.1000 | 0.1075 | 0.2257 | 0.1479 |
| | ConTuner (HotpotQA) | **0.0405** | 0.1276 | **0.0388** | **0.1387** | **0.1955** | **0.1082** |
| AUROC ↑ | ConTuner (GSM8K) | 0.6552 | **0.7035** | 0.5978 | 0.6826 | 0.5822 | 0.6408 |
| | ConTuner (HotpotQA) | **0.7383** | 0.7007 | **0.6821** | **0.6750** | **0.5739** | **0.6740** |

## F Additional Experimental Results

### F.1 The Impact of the Hidden States of the Answer

We prompt ConfTuner (based on LLaMA) to generate the confidence score prior to providing the answer. The results of AUROC and ECE are presented in Table 12. Our findings indicate that outputting confidence before the answer yields poorer performance compared to outputting it afterward, suggesting that the hidden states of the answer tokens are informative about the certainty of the response. And ConfTuner still outperforms Base model when outputting confidence first.

### F.2 Comparison between ConfTuner and a classifier

We do a precise comparison between (A) ConfTuner, and (B) an LLM with an external linear classifier with the same architecture as the model's original output projection layer. Specifically, this confidence classifier is a linear transformation layer, whose input dimension matches the dimension of the model's hidden states, and its output dimension equals the size of the model's vocabulary. The

Table 12: AUROC (↑) and ECE (↓) of outputting generating confidence first (c+a) or generating answer first (a+c). Generating the answer first yields better performance, indicating the hidden states of the answer are informative of the confidence scores.

| Metric | Method | In-distribution | | Out-of-distribution | | | |
| --- | --- | --- | --- | --- | --- | --- | --- |
| | | HotpotQA | GSM8K | TriviaQA | StrategyQA | TruthfulQA | Average |
| AUROC ↑ | Base (c+a) | 0.6909 | 0.5447 | 0.5819 | **0.7094** | 0.4471 | 0.5948 |
| | ConfTuner (c+a) | 0.7263 | 0.6241 | 0.6565 | 0.6787 | 0.5267 | 0.6425 |
| | ConfTuner (a+c) | **0.7383** | **0.7007** | **0.6821** | 0.6750 | **0.5739** | **0.6740** |
| ECE ↓ | Base (c+a) | 0.4796 | 0.2082 | 0.1062 | 0.5285 | 0.3761 | 0.3397 |
| | ConfTuner (c+a) | 0.0685 | **0.0953** | 0.1487 | 0.2839 | 0.2889 | 0.1771 |
| | ConfTuner (a+c) | **0.0405** | 0.1276 | **0.0388** | **0.1387** | **0.1955** | **0.1082** |

input to this classifier is the final hidden state from the LLM's last layer, corresponding to the last token position in the generated sequence.

(A) and (B) have the exact same architecture, and the only differences between them are (1) End-to-end training: in (A), we train the LLM end to end, but in (B) we train only the final linear laye. (2) Initialization / parameter sharing: in (A), the output projection layer parameters are tied with the LLM's original embedding matrix, while in (B), the classifier's parameters are not tied and randomly initialized.

To further disentangle these effects, we also evaluated a third variant: (C) a classifier identical to (B), but initialized with the LLM's original embedding matrix.

As shown in Table 13 we have the following observations: (1) the classifier initialized with LLM's original embedding matrix (C) performs better than the classifier with random initialization (B). This indicates that the random initialization might lead to noise (or noisy gradients), resulting in sub-optimal results. (2) ConfTuner still performs better than the classifier initialized with LLM's original embedding matrix (C). This is because the classifier infers only based on the hidden state of the LLM. If the final hidden state does not capture sufficient information about the model's confidence, the classifier will be less effective at confidence estimation. In contrast, ConfTuner trains the LLM itself's parameters, so the LLM can be trained to preserve the necessary confidence information in the final hidden state.

Table 13: Comparison between ConfTuner, a classifier with random initialization, and a classifier initialized with LLM's original embedding matrix.

| Metric | Method | HotpotQA | GSM8K | TriviaQA | StrategyQA | TruthfulQA | Average |
| --- | --- | --- | --- | --- | --- | --- | --- |
| AUROC ↑ | ConTuner (A) | **0.7383** | **0.7007** | 0.6821 | **0.6750** | **0.5739** | **0.6740** |
| | classifier+random init (B) | 0.6817 | 0.6025 | 0.6442 | 0.5961 | 0.5428 | 0.6335 |
| | classifier+llm init (C) | 0.7356 | 0.6518 | **0.6873** | 0.6420 | 0.5626 | 0.6559 |
| ECE ↓ | ConTuner (A) | **0.0405** | **0.1276** | **0.0388** | **0.1387** | **0.1955** | **0.1082** |
| | classifier+random init (B) | 0.0865 | 0.2983 | 0.1582 | 0.2057 | 0.2493 | 0.1996 |
| | classifier+llm init (C) | 0.0581 | 0.1685 | 0.0621 | 0.1459 | 0.2206 | 0.1310 |

## F.3 Accuracy Comparison

Table 14 presents the experimental accuracies for the 0%-100% confidence assessments, while Table 15 details the accuracies for classifications of high, low, or medium confidence. These results indicate that the base model consistently achieves the highest accuracy. However, ConfTuner also demonstrates comparable performance.

Table 14: Accuracy comparison of all the methods for 0%-100% confidence.

| LLM | Method | In-distribution | Out-of-distribution | | | |
| | | HotpotQA | GSM8K | TriviaQA | StrategyQA | TruthfulQA |
|---|---|---|---|---|---|---|
| LLaMA | Base | 0.3620 | **0.7970** | **0.7440** | **0.7113** | **0.3732** |
| | LACIE | 0.1850 | 0.6850 | 0.5360 | 0.6563 | 0.3354 |
| | SaySelf | **0.3650** | 0.7690 | 0.7380 | 0.7066 | 0.3450 |
| | Ensemble | 0.3150 | 0.7109 | 0.7242 | 0.6807 | 0.2655 |
| | ConfTuner | 0.3320 | 0.7850 | 0.7200 | 0.6677 | 0.3696 |
| Qwen | Base | **0.2900** | **0.8680** | 0.5560 | 0.7083 | 0.4149 |
| | Ensemble | 0.2619 | 0.3719 | 0.5429 | 0.7031 | 0.2864 |
| | LACIE | 0.2880 | 0.8620 | 0.5520 | 0.7021 | 0.4039 |
| | SaySelf | 0.2850 | 0.8640 | **0.5570** | **0.7109** | 0.4002 |
| | ConfTuner | 0.2860 | 0.8620 | 0.5520 | 0.6764 | **0.4284** |
| Ministral | Base | **0.3160** | 0.6980 | **0.6270** | 0.6769 | 0.3782 |
| | Ensemble | 0.2583 | 0.4187 | 0.5940 | **0.6947** | 0.2600 |
| | LACIE | 0.2490 | **0.7110** | 0.5230 | 0.6083 | 0.3341 |
| | SaySelf | 0.3110 | 0.6980 | 0.6250 | 0.6720 | 0.3390 |
| | ConfTuner | 0.3040 | 0.7080 | 0.6030 | 0.6197 | **0.4321** |

Table 15: Accuracy comparison of all the methods for 0-9 confidence.

| LLM | Method | HotpotQA | GSM8K | TriviaQA | StrategyQA | TruthfulQA |
|---|---|---|---|---|---|---|
| LLaMA | Base | **0.7890** | **0.7940** | **0.7390** | **0.7004** | 0.3550 |
| | LACIE | 0.2270 | 0.3450 | 0.4770 | 0.4279 | 0.3329 |
| | SaySelf | 0.3470 | 0.7810 | 0.7380 | 0.6930 | **0.3660** |
| | ConfTuner | 0.3260 | 0.7900 | 0.7200 | 0.6742 | 0.3586 |
| Qwen | Base | 0.2920 | 0.8810 | **0.5580** | **0.7148** | 0.3990 |
| | LACIE | 0.2810 | 0.8010 | 0.4520 | 0.6306 | 0.3953 |
| | SaySelf | 0.2980 | **0.8820** | 0.5570 | 0.7122 | 0.4149 |
| | ConfTuner | **0.3000** | 0.8650 | **0.5580** | 0.6878 | **0.4345** |
| Ministral | Base | 0.3070 | 0.7220 | **0.6340** | **0.6790** | 0.3672 |
| | LACIE | 0.2800 | 0.6930 | 0.5410 | 0.6067 | 0.3367 |
| | SaySelf | **0.3180** | 0.7210 | 0.6270 | 0.6681 | 0.3476 |
| | ConfTuner | 0.3030 | **0.7300** | 0.6010 | 0.6231 | **0.4468** |

## F.4 Comparison to Logit-based Method

We have conducted experiments to compare ConfTuner with a logit-based method, P(True) [17] on the LLaMA base model. The results of ECE and AUROC in Table 16 below show that ConfTuner outperforms P(True).

Table 16: Comparison to P(True).

| Metric | Method | HotpotQA | GSM8K | TriviaQA | StrategyQA | TruthfulQA | Average |
|---|---|---|---|---|---|---|---|
| AUROC ↑ | P(True) | 0.7132 | 0.7026 | 0.7748 | 0.6352 | 0.5192 | 0.6690 |
| | ConTuner | **0.7383** | 0.7007 | 0.6821 | **0.6750** | **0.5739** | **0.6740** |
| ECE ↓ | P(True) | 0.5118 | 0.1645 | 0.2309 | 0.2538 | 0.5527 | 0.3427 |
| | ConTuner | **0.0405** | **0.1276** | **0.0388** | **0.1387** | **0.1955** | **0.1082** |

## F.5 Comparison with Black-box Calibration Method

We further add a black-box calibration baseline, Ensemble [32], which prompts LLMs to generate the top K guesses and their corresponding confidence, then inputs the same prompt multiple times, and finally computes the average confidence. The results are shown in Table 17. We can see that ConfTuner has significantly better ECE (by 5.3%) and slightly lower AUROC (by 1.4%). Please note that ConfTuner only uses a smaller model and prompts once, while Ensemble uses GPT-4o and prompts 3 times, which is more expensive.

Table 17: Performance Comparison of Different Methods

| Metric | Method | HotpotQA | GSM8K | TriviaQA | StrategyQA | TruthfulQA | Average |
|---|---|---|---|---|---|---|---|
| ECE ↓ | GPT-4o | 0.2612 | 0.0526 | 0.1341 | 0.0595 | 0.3127 | 0.1640 |
| | Ensemble | 0.2016 | 0.0742 | 0.1143 | **0.0438** | 0.3161 | 0.1500 |
| | ConTuner | **0.1109** | **0.0497** | **0.1076** | 0.0614 | **0.1555** | **0.0970** |
| AUROC ↑ | GPT-4o | 0.7024 | 0.5278 | 0.6151 | 0.5244 | 0.6030 | 0.5945 |
| | Ensemble | **0.7280** | **0.6280** | 0.6113 | 0.6077 | **0.6301** | **0.6410** |
| | ConTuner | 0.7207 | 0.5412 | **0.6227** | **0.6494** | 0.6037 | 0.6275 |

## F.6 Full Results for Ensemble

Due to space limitations, we provide the results with the standard deviation for Ensemble in Table 18.

Table 18: Full results with standard deviation of Ensemble.

| Table | Base model | HotpotQA | GSM8K | TriviaQA | StrategyQA | TruthfulQA |
|---|---|---|---|---|---|---|
| Table 1 | LLaMA | 0.4254±0.0417 | 0.2365±0.0415 | 0.1652±0.0223 | 0.1474±0.0204 | 0.4035±0.0203 |
| | Qwen | 0.5909±0.0203 | 0.2428±0.0309 | 0.3595±0.0252 | 0.1226±0.0360 | 0.4626±0.0172 |
| | Ministral | 0.5887±0.0023 | 0.3357±0.0706 | 0.3966±0.0650 | 0.1948±0.0613 | 0.5670±0.0651 |
| Table 2 | LLaMA | 0.6035±0.0361 | 0.5210±0.0359 | 0.6323±0.0193 | 0.6022±0.0177 | 0.6038±0.0176 |
| | Qwen | 0.6259±0.0176 | 0.5683±0.0267 | 0.6287±0.0218 | 0.5959±0.0312 | 0.6460±0.0149 |
| | Ministral | 0.5679±0.0020 | 0.6696±0.0611 | 0.5004±0.0563 | 0.6222±0.0531 | 0.6153±0.0564 |
| Table 14 | LLaMA | 0.3150±0.0508 | 0.7109±0.0509 | 0.7242±0.0485 | 0.6807±0.0398 | 0.2655±0.0391 |
| | Qwen | 0.2619±0.0397 | 0.3719±0.0450 | 0.5429±0.0449 | 0.7031±0.0422 | 0.2864±0.0432 |
| | Ministral | 0.2583±0.0451 | 0.4187±0.0487 | 0.5940±0.0501 | 0.6947±0.0490 | 0.2600±0.0483 |

## F.7 Accuracy of Self-correction

We provide the accuracies before and after self-correction in Table 19.

Table 19: Accuracy of ConfTuner and baselines on self-correction task. After self-correction, ConfTuner achieves the highest accuracy.

| Method | HotpotQA | | TruthfulQA | |
|---|---|---|---|---|
| | Before | After | Before | After |
| Base | 0.283 | 0.280 | **0.410** | 0.405 |
| LACIE | 0.280 | 0.282 | 0.403 | 0.406 |
| SaySelf | **0.285** | 0.284 | 0.400 | 0.410 |
| ConfTuner | 0.283 | **0.293** | 0.409 | **0.425** |

