# OpenReview forum: "ConfTuner: Training Large Language Models to Express Their Confidence Verbally"
_NeurIPS.cc/2025/Conference — NeurIPS 2025 poster_

### Official Review · Reviewer_9LBm · 2025-06-19

**Clarity:** 3
**Significance:** 3
**Originality:** 2
**Rating:** 4
**Confidence:** 4

**Summary:**

The paper that I read is about a significant issue with Large Language Models (LLMs) that frequently overestimate their confidence in their responses. To tackle this problem, the researchers came up with ConfTuner, a brand-new technique for fine-tuning LLMs that enables them to communicate their confidence levels in a well-calibrated manner. They created a "tokenized Brier score" loss function that encourages the model to express confidence levels that accurately reflect its likelihood of being correct. The best part is that ConfTuner doesn't require any ground-truth confidence scores and only adds a little computational overhead. The researchers tested ConfTuner on various reasoning tasks and discovered that it outperforms existing calibration techniques and can even be used with black-box models like GPT-4o.

**Questions:**

1. If we want to elicit confidence $\in \mathbb{R}$, why not add a regression head after the llm generated answer and train that with Brier score? I would suggest to ablate against a regression head baseline (e.g., attach a scalar head to the LLM’s final hidden state, trained with Brier loss) to establish the superiority of tokenised confidence.

2. It would be valuable to evaluate how the proposed method performs relative to black-box models where confidence estimation is derived directly from access to output logits.

3. In Experiment section please evaluate against Pair-rank to aggregate topK sampled verbalized confidence scores - https://openreview.net/forum?id=gjeQKFxFpZ

4. The sensitivity of the confidence estimates to varying prompting strategies remains unclear. It would be useful to quantify the incremental gains from fine-tuning, compared to improved prompt engineering and top-k response aggregation using existing models.

5. Limited Ablation on Training Distribution Shifts: The paper could benefit from more extensive analysis on how shifts in the training data distribution affect calibration performance.

**Ethical Concerns:**

["NO or VERY MINOR ethics concerns only"]

**Final Justification:**

The authors have addressed most of my concerns, and I have accordingly revised my score. However, the novelty of the work remains unclear to me. The use of Brier score-based loss functions has been well established in the uncertainty estimation literature for deep neural networks. As a result, this work comes across more as a straightforward extension of those ideas to the LLM setting using a simplex-based approach.

**Limitations:**

The authors acknowledge the limitation of tokenization rigidity, where the use of predefined tokens may constrain flexibility. However, they do not highlight data dependency as a critical factor—calibration quality is highly sensitive to the training data distribution. Performance may degrade when there are shifts in topic or style, such as transitioning from medical to legal QA.

Additionally, the authors did not address the fairness implications of their approach. There is no analysis on whether calibration quality varies across demographic groups or in cases of ambiguous queries (e.g., non-Western names in TriviaQA), which may introduce biases.

**Paper Formatting Concerns:**

None.

**Quality:**

3

**Strengths And Weaknesses:**

Strength:
1. Novel Formulation of Verbalized Calibration: Adapts classical proper scoring rules (e.g., Brier score) to tokenized confidence expressions, bridging a theoretical gap between traditional classifiers and LLMs.
2. Practical Efficiency: Minimal overhead: Trains with 2,000 samples (vs. 100k for baselines) and converges in 4 minutes (vs. 26–120 min). Requires no ground-truth confidence or repeated sampling, making it scalable.

Weakness:
1. Limited Exploration of Complex Contexts: While the paper shows generalization to different confidence formats (numerical percentages, high/medium/low), it doesn't fully explore more complex conversational contexts or diverse natural language expressions of uncertainty.
2. Black-Box Calibration Limitations: While ConfTuner calibrates GPT-4o outputs, it relies on a proxy model (LLaMA) and cannot access internal states. Experiments lack comparison to other black-box calibration methods.

---

> ### Author Rebuttal · Authors · 2025-07-31
>
> Thanks for your valuable suggestions! We address your main concerns as follows:
>
> > ### W1: Limited exploration of complex contexts
>
> **ConfTuner can generalize to more complex contexts**. We conduct tests with a more **open-ended setup**. Specifically, we change the original prompt from `{Question + output confidence (0-100)}` to `{Question + Please express your uncertainty when providing the answer}`. This new setting significantly expands the LLM's output space to diverse natural language expressions of uncertainty, such as "I'm fairly certain, but ..." or "I'm not sure" We evaluated these implicit confidences by inputting them to GPT-4o to assess the implied confidence levels. The results of AUROC and ECE are presented in Tables 1 and 2 below,  showcasing ConfTuner's robustness in diverse conversational contexts.
>
> Table 1: AUROC
>
> | Method | HotpotQA | GSM8K | TriviaQA | StrategyQA | TruthfulQA | Average |
> |-|-|-|-|-|-|-|
> | Base (implicit) | 0.7047 | 0.5422 | 0.6342 | 0.6489 | 0.5895 | 0.6239 |
> | ConfTuner (explicit) | **0.7383** | **0.7007** | 0.6821 | 0.6750 | 0.5739 | 0.6740 |
> | ConfTuner (implicit) | 0.7239 | 0.6869 | **0.7024** | **0.6751** | **0.6217** | **0.6820** |
>
> Table 2: ECE
>
> | Method | HotpotQA | GSM8K | TriviaQA | StrategyQA | TruthfulQA | Average |
> |-|-|-|-|-|-|-|
> | Base (implicit) | 0.2808 | 0.1179 | 0.1232 | **0.1098** | 0.3250 | 0.1913 |
> | ConfTuner (explicit) | **0.0405** | 0.1276 | **0.0388** | 0.1387 | **0.1955** | **0.1082** |
> | ConfTuner (implicit) | 0.1639 | **0.0950** | 0.1088 | 0.1721 | 0.2019 | 0.1483 |
>
> > ### W2: Comparison of black-box calibration methods for GPT-4o.
>
> Based on your suggestion, **we further add a black-box calibration baseline, Ensemble [1]**, which prompts LLMs to generate the top K guesses and their corresponding confidence, then inputs the same prompt multiple times, and finally computes the average confidence. The results are shown in Tables 3 and 4. We can see that **ConfTuner has significantly better ECE (by 5.3%) and slightly lower AUROC (by 1.4%)**. Please note that ConfTuner only uses a smaller model and prompts once, while Ensemble uses GPT-4o and prompts 3 times, which is more expensive.
>
> Table 3: ECE
>
>
> | Method | HotpotQA | GSM8K | TriviaQA | StrategyQA | TruthfulQA | Average |
> |-|-|-|-|-|-|-|
> | GPT4o | 0.2612 | 0.0526 | 0.1341 | 0.0595 | 0.3127 | 0.1640 |
> | Ensemble | 0.2016 | 0.0742 | 0.1143 | **0.0438** | 0.3161 | 0.1500 |
> | ConfTuner | **0.1109** | **0.0497** | **0.1076** | 0.0614 | **0.1555** | **0.0970** |
>
> Table 4: AUROC
>
> | Method | HotpotQA | GSM8K | TriviaQA | StrategyQA | TruthfulQA | Average |
> |-|-|-|-|-|-|-|
> | GPT-4o | 0.7024 | 0.5278 | 0.6151 | 0.5244 | 0.6030 | 0.5945 |
> | Ensemble | **0.7280** | **0.6280** | 0.6113 | 0.6077 | **0.6301** | **0.6410** |
> | ConfTuner | 0.7207 | 0.5412 | **0.6227** | **0.6494** | 0.6037 | 0.6275 |
>
> > ### Q1: Ablate against a regression head baseline
>
> ConfTuner has two strengths over a regression head:  1) **ConfTuner enhances the model's intrinsic calibration ability, thus has better generalization ability**. ConfTuner could generalize to out-of-distribution datasets, different forms of confidence levels (Section 4.2), and to **diverse and more complex confidence expressions (W1), which cannot be done by a simple regression head**. 2) It offers a **more seamless and natural** way to express confidence.
>
> Brier loss is used for various confidence level tokens instead of a single numeric value, so we further compare the performance of ConfTuner and a classifier on the in-distribution and out-of-distribution datasets. We conducted an experiment where we fed the token logits after "Confidence: " from the last hidden state into a two-layer MLP classifier to predict confidence scores from 0 to 100, trained with Brier loss. The results, presented in Tables 5 and 6 below, compare this approach to ConfTuner. Our findings indicate that **the classifier-based method yields poorer performance compared to ConfTuner, especially on OOD datasets**.
>
> Table 5: AUROC
>
> | Method | HotpotQA | GSM8K | TriviaQA | StrategyQA | TruthfulQA | Average |
> |-|-|-|-|-|-|-|
> | Classifier | 0.7340 | 0.5843 | 0.6425 | 0.6219 | 0.5648 |  0.6295 |
> | ConfTuner | **0.7383** | **0.7007** | **0.6821** | **0.6750** | **0.5739** | **0.6740** |
>
> Table 6: ECE
>
> | Method | HotpotQA | GSM8K | TriviaQA | StrategyQA | TruthfulQA | Average |
> |-|-|-|-|-|-|-|
> | Classifier | 0.1253 | 0.2261 | 0.1469 | 0.1526 | 0.3894 |  0.2081 |
> | ConfTuner | **0.0405** | **0.1276** | **0.0388** | **0.1387** | **0.1955** | **0.1082** |
>
> > ### Q2: Comparison to models where confidence estimation is derived from logits
>
> We further use a method [3], P(True), where confidence estimation is derived directly from logits of the token “True”. The results of ECE and AUROC are shown in Tables 7 and 8, demonstrating that **ConfTuner performs better than P(True)**.
>
> Table 7:  AUROC
>
> | Method | HotpotQA | GSM8K | TriviaQA | StrategyQA | TruthfulQA | Average |
> |-|-|-|-|-|-|-|
> | P(True) | 0.7132 | **0.7026** | **0.7748** | 0.6352 | 0.5192 | 0.6690 |
> | ConfTuner | **0.7383** | 0.7007 | 0.6821 | **0.6750** | **0.5739** | **0.6740** |
>
> Table 8: ECE
>
> | Method | HotpotQA | GSM8K | TriviaQA | StrategyQA | TruthfulQA | Average |
> |-|-|-|-|-|-|-|
> | P(True) | 0.5118 | 0.1645 | 0.2309 | 0.2538 | 0.5527 | 0.3427 |
> | ConfTuner | **0.0405** | **0.1276** | **0.0388** | **0.1387** | **0.1955** | **0.1082** |
>
>
> > ### Q3: Comparison to Ensemble (pair-rank).
>
> We provide the results of Pair-rank in Tables 9 and 10. The results demonstrate that **ConfTuner outperforms Ensemble (Pair-rank)**.
>
> Table 9: ECE
>
> | Method | HotpotQA | GSM8K | TriviaQA | StrategyQA | TruthfulQA | Average |
> |-|-|-|-|-|-|-|
> | Ensemble(Pair-rank) | 0.4745 | 0.2161 | 0.1535 | 0.1862 | 0.4165 | 0.2894 |
> | ConfTuner | **0.0405** | **0.1276** | **0.0388** | **0.1387** | **0.1955** | **0.1082** |
>
> Table 10: AUROC
>
> | Method | HotpotQA | GSM8K | TriviaQA | StrategyQA | TruthfulQA | Average |
> |-|-|-|-|-|-|-|
> | Ensemble(Pair-rank) | 0.6237 | 0.5768 | 0.6411 | 0.6388 | **0.6096** | 0.6180 |
> | ConfTuner | **0.7383** | **0.7007** | **0.6821** | **0.6750** | 0.5739 | **0.6740** |
>
> > ### Q4: Sensitivity of ConfTuner to varying prompting strategies & Comparison to improved prompt engineering and top-k response aggregation
>
> Even trained only on numeric confidence (0-100%) prompting, **ConfTuner could generalize to different confidence elicitation prompting strategies**, such as "high/medium/low" (Table 3 in our paper), and implicit confidence elicitation (W1).
>
> We have compared ConfTuner with Ensemble (Tables 1 and 2 in our paper), where the prompting and aggregation strategies are recommended by [1]. We also compare ConfTuner with Ensemble (pair-rank) in Q3. The results show that ConfTuner still achieves better performance.
>
> We further conduct an experiment on ConfTuner + Ensemble, i.e., use the prompting and aggregation strategies of Ensemble for ConfTuner. The results in Tables 11 and 12 show that **improved prompting and aggregation strategies could yield better performance for ConfTuner**.
>
> Table 11: ECE
>
> | Method | HotpotQA | GSM8K | TriviaQA | StrategyQA | TruthfulQA | Average |
> |-|-|-|-|-|-|-|
> | ConfTuner + Ensemble | 0.0518 |**0.0824** | 0.0567 | **0.0736** | **0.1629**  | **0.0855**  |
> | ConfTuner | **0.0405** | 0.1276 |**0.0388**| 0.1387 | 0.1955 | 0.1082 |
>
> Table 12: AUROC
>
> | Method | HotpotQA | GSM8K | TriviaQA | StrategyQA | TruthfulQA | Average |
> |-|-|-|-|-|-|-|
> | ConfTuner + Ensemble| 0.7264 | **0.7151**| **0.7079** | **0.6825**  | **0.5973** |   **0.6858**|
> | ConfTuner | **0.7383** | 0.7007 | 0.6821 | 0.6750 | 0.5739 | 0.6740 |
>
>
> > ### Q5 and Limitation 1: Ablation on Training Distribution Shifts
>
> We further train LLaMA on GSM8K (math problems) instead of HotpotQA (general knowledge from Wikipedia). As shown in Tables 13 and 14. **ConfTuner trained on GSM8K performs better on GSM8K and StrategyQA, but worse on HotpotQA, TriviaQA, and TruthfulQA**.
>
> Table 13: ECE
>
> | Method | HotpotQA | GSM8K | TriviaQA | StrategyQA | TruthfulQA | Average |
> |-|-|-|-|-|-|-|
> | ConfTuner (GSM8K) | 0.2308 | **0.0753** | 0.1000 | **0.1075** | 0.2257 | 0.1479 |
> | ConfTuner (HotpotQA) | **0.0405** | 0.1276 | **0.0388** | 0.1387 | **0.1955** | **0.1082** |
>
> Table 14: AUROC
>
> | Method | HotpotQA | GSM8K | TriviaQA | StrategyQA | TruthfulQA | Average |
> |-|-|-|-|-|-|-|
> | ConfTuner (GSM8K) | 0.6552 | **0.7035** | 0.5978 | **0.6826** | **0.5822** | 0.6408 |
> | ConfTuner (HotpotQA) | **0.7383** | 0.7007 | **0.6821** | 0.6750 | 0.5739 | **0.6740** |
>
> > ### Limitation 2: fairness implications of ConfTuner
>
> The primary focus of this paper is to establish effective verbalized confidence calibration. Analyzing calibration quality across demographic groups or ambiguous queries is a critical topic and a valuable direction for future work. Indeed, [2] finds that many existing calibrators tend to focus on calibrating overall performance, which can lead to miscalibration in underrepresented groups. We believe that methods that more deeply integrate with the LLM’s own capabilities like ours have potential to alleviate this problem, but a deeper analysis is left for future work. We appreciate your suggestion and will include a discussion on fairness implications and potential biases in our revised manuscript.
>
> ​​Thanks again for your constructive and valuable suggestions! We'll incorporate them in future revisions.​​ Please let us know if you have any further questions.
>
> [1] Can llms express their uncertainty? an empirical evaluation of confidence elicitation in llms
>
> [2] Proximity-Informed Calibration for Deep Neural Networks.
>
> [3] Language Models (Mostly) Know What They Know

---

### Official Review · Reviewer_wrZQ · 2025-06-28

**Clarity:** 3
**Significance:** 3
**Originality:** 3
**Rating:** 4
**Confidence:** 4

**Summary:**

The authors argue for a loss based on a proper scoring rule for improving the calibrations of LLMs. They tackle the challenge of verbalizing the calibrated confidence of LLMs and towards this they propose a method called ConfTuner which finetunes an LLM on correctness scores to elicit the confidence scores. While a vanilla application of a proper scoring rule based loss function, such as Brier score, would produce a numerical score, the challenge is how to verbalize this numerical score? Towards this end, the author propose a loss function, inspired by a Brier scoring rule, that assigns high probability to low-confidence (verbalized) tokens (and low probability to high-confidence tokens) when the LLM is wrong and the opposite when the LLM is right. When an LLM is finetuned with such a loss function based on correctness, it produces calibrated confidence scores. Experimental results suggest that the generated confidence scores are well-calibrated on the dataset for which finetuning set was used but calibration is significantly worse for out-of-distribution datasets (for which finetuning was not done).

**Questions:**

See the weaknesses section above.

**Ethical Concerns:**

["NO or VERY MINOR ethics concerns only"]

**Final Justification:**

My post-rebuttal response
-----------------------------

Thank you for providing clarification and adding the P(True) baseline. The AUROC is almost the same as ConfTuner. I keep my original score.

**Limitations:**

See the weaknesses section above.

**Paper Formatting Concerns:**

None.

**Quality:**

3

**Strengths And Weaknesses:**

**Strengths**:
- The paper is well-written and easy to follow. The proposed method is well-motivated.
- The Brier score inspired loss function is neat and seems to generate well-calibrated scores for in-distribution examples.

** Weaknesses **:
- The main challenge with finetuning-based calibration approaches is the difficulty in demonstrating that the model is not merely exploiting spurious correlations in (question, answer, confidence) tuples to to achieve low loss on a given dataset, and that it is truly learning to generate a calibrated confidence scores. This clearly manifests in the lackluster out-of-distribution results on many fine-tuning-based calibration methods. I am afraid this work has similarly quite high out-of-distribution calibration errors. Could the authors comment on that and how will they rebut the concern that their method is not merely exploiting the distributional similarity in (question, answer, confidence) tuples for the in-distribution examples and not really learning to emit a calibrated confidence score.

- I think an important baseline of self-evaluation task (proposed in Kadavath et al. (https://arxiv.org/abs/2207.05221) is missing. Please add that.

---

> ### Author Rebuttal · Authors · 2025-07-31
>
> Thanks for your positive feedback and valuable suggestions! We address your main concerns as follows:
>
> > ### W1: Concerns of exploiting spurious correlations in (question, answer, confidence) tuples and high out-of-distribution calibration errors.
>
> The absolute levels of metrics on a given dataset depend significantly on the dataset’s difficulty and the model’s accuracy on that task, which vary from one dataset to another. However, in many cases, **the improvements contributed by our approach over the Base model on out-of-distribution cases are often better than on in-distribution ones**. The results are shown in Table 1 below. Under LLaMA and Qwen models, the AUROC improvements on OOD datasets generally surpass those on the in-distribution dataset (HotpotQA). This suggests that ConfTuner is not merely exploiting distributional similarities in the training data but is learning robust confidence calibration capabilities.
>
> ConfTuner not only generalizes to OOD datasets, but also **generalizes to different forms of confidence expressions** (e.g., training on confidence expressed as percentages, and then expressing confidence linguistically as high/medium/low at test time), as demonstrated in Section 4.2. This reinforces that ConfTuner has internalized calibration ability rather than relying on spurious correlations.
>
> Furthermore, we conduct a new experiment to show that **ConfTuner also generalizes to implicit confidence expressions**. Specifically, we modified the original prompt from `{Question + output confidence (0-100)}` to `{Question + Please express your uncertainty when providing the answer}`. This new setting significantly expands the LLM's output space, allowing it to generate diverse natural language expressions of uncertainty, such as "I'm fairly certain, but there’s a chance I could be mistaken" or "This is a tough one, so I’d say it’s likely but not guaranteed." We evaluated these implicit confidences by inputting them to GPT-4o to assess the implied confidence levels. The results of AUROC and ECE are presented in Tables 2 and 3 below, demonstrating that **ConfTuner enhances the model's intrinsic calibration ability instead of simply learning spurious correlations**.
>
>
> Table 1: Improvement of AUROC compared to the Base model
>
> | LLM | HotpotQA (In-dist) | GSM8K | TriviaQA | StrategyQA | TruthfulQA |
> |------|-------------------|-------|----------|------------|------------|
> | LLaMA | +7.25% | +39.35% | +13.25% | +8.02% | +5.63% |
> | Qwen | +4.62% | +14.22% | +23.13% | +10.45% | +6.28% |
> | Ministral | +52.12% | +30.53% | +45.51% | +0.35% | +24.64% |
>
>
> Table 2: Implicit confidence expression performance (AUROC)
> | Method | HotpotQA | GSM8K | TriviaQA | StrategyQA | TruthfulQA | Average |
> |--------|----------|--------|-----------|------------|------------|---------|
> | Base (implicit) | 0.7047 | 0.5422 | 0.6342 | 0.6489 | 0.5895 | 0.6239 |
> | ConfTuner (explicit) | **0.7383** | **0.7007** | 0.6821 | 0.6750 | 0.5739 | 0.6740 |
> | ConfTuner (implicit) | 0.7239 | 0.6869 | **0.7024** | **0.6751** | **0.6217** | **0.6820** |
>
> Table 3: Implicit confidence expression performance (ECE)
>
> | Method | HotpotQA | GSM8K | TriviaQA | StrategyQA | TruthfulQA | Average |
> |--------|----------|--------|-----------|------------|------------|---------|
> | Base (implicit) | 0.2808 | 0.1179 | 0.1232 | **0.1098** | 0.3250 | 0.1913 |
> | ConfTuner (explicit) | **0.0405** | 0.1276 | **0.0388** | 0.1387 | **0.1955** | **0.1082** |
> | ConfTuner (implicit) | 0.1639 | **0.0950** | 0.1088 | 0.1721 | 0.2019 | 0.1483 |
>
> > ### W2: Comparison to a related work.
>
> Our method focuses on verbalized confidence calibration and differs from the work [1], which uses logits to infer the confidence. Thus, we initially did not include it as a baseline. However, per your request, we have conducted experiments incorporating this baseline (P(True)) on the LLaMA base model. The results of ECE and AUROC in Tables 4 and 5 below show that **ConfTuner outperforms P(True)** [1]. We will add this baseline to our revised manuscript.
>
> Table 4: Comparison to P(True) (AUROC)
>
> | Method | HotpotQA | GSM8K | TriviaQA | StrategyQA | TruthfulQA | Average |
> |--------|----------|-------|----------|------------|------------|---------|
> | P(True) | 0.7132 | **0.7026** | **0.7748** | 0.6352 | 0.5192 | 0.6690 |
> | ConfTuner | **0.7383** | 0.7007 | 0.6821 | **0.6750** | **0.5739** | **0.6740** |
>
>
> Table 5: Comparison to P(True) (ECE)
>
> | Method | HotpotQA | GSM8K | TriviaQA | StrategyQA | TruthfulQA | Average |
> |--------|----------|-------|----------|------------|------------|---------|
> | P(True) | 0.5118 | 0.1645 | 0.2309 | 0.2538 | 0.5527 | 0.3427 |
> | ConfTuner | **0.0405** | **0.1276** | **0.0388** | **0.1387** | **0.1955** | **0.1082** |
>
>
> We really appreciate your valuable suggestions; they’ll be extremely useful for our future work on revising the manuscript.​​ Please let us know if you have any further questions.
>
> [1] Language Models (Mostly) Know What They Know

---

> > ### Comment · Reviewer_wrZQ · 2025-08-04
> > **Post-rebuttal response**
> >
> > Thank you for providing clarification and adding the P(True) baseline. The AUROC is almost the same as ConfTuner. I keep my original score.

---

> > > ### Author Response · Authors · 2025-08-05
> > > **Response to Reviewer wrZQ**
> > >
> > > Thank you for your thorough review and positive feedback. As suggested, we will integrate these experimental findings and the corresponding discussion into our final revision. Your time and expertise were instrumental in strengthening this work.

---

### Official Review · Reviewer_Fgyi · 2025-06-28

**Clarity:** 4
**Significance:** 3
**Originality:** 3
**Rating:** 5
**Confidence:** 4

**Summary:**

The paper proposes to "verbalize" (un)certainty in LLM answers by prepending "Confidence: <>" to the answer, where in place of "<>", the scores for the token predictions (e.g. 0,1,2,...99,100) are evaluated as the uncertainty estimate. The confidence token predictions are trained using a tokenized Brier score inspired by proper scoring results. The method is evaluated on several standard benchmarks and report strong results, outperforming evaluated baselines.

**Questions:**

1. How load bearing is the "verbalization" part here? Effectively, we are using the output embedding matrix as a 101-class classifier between 0...100. Would training a basic classifier on top of the last hidden state after appending "Confidence: " to the generated answer work just as well? Or is actually using the pretrained "verbalized" representation of the embedding tokens 0...100 crucial? --- More generally, what are the actual differences between your proposed method and a "simple" classification into confidence bins? A proper clarification on this point will alleviate my concerns regarding originality.

2. Would outputting the confidence prediction *before* the answer change how well the method works? In other words, are the hidden states of the actual answer tokens informative about the certainty of the answer or is this mostly extracted from hidden states in the question? This might be a nice additional experiment but not related a major concern regarding the paper.

**Ethical Concerns:**

["NO or VERY MINOR ethics concerns only"]

**Final Justification:**

I am raising the score of my review. The authors have meaningfully engaged with the Reviewers and conducted significant, insightful additional experiments. Additionally, the discussion has been fruitful; the authors have candidly discussed limitations of their method and ablated various components to give a better assessment of the mechanisms behind their method. If the authors add the additional experiments and points of discussion to the paper, it turns an already good work into a strong one. While not groundbreaking for AI as a whole, the paper in that state would be a technically well-executed and meaningful addition to the current research area of uncertainty quantification in LLMs.

Therefore I raise the score to 5 — `Accept`.

**Limitations:**

yes

**Quality:**

4

**Strengths And Weaknesses:**

### Strengths
The method is well-motivated and the introduction of the tokenized Brier score is novel. The authors also include valuable experiments on self-correction and model cascades as evaluations how uncertainty estimation methods can be employed in practice to improve the performance of LLM (systems).

### Weaknesses
Fundamentally, I do not see the difference between the proposed method and a classification problem into confidence bins (which itself can be seen as a discretization of a regression task predicting the confidence) --- See also my question. Therefore, while the method itself is well-motivated and reports string results, I would like to see more discussion on this point or arguments to the contrary.

There is also an abundant amount of research into uncertainty quantification nowadays (which I will acknowledge is impractically to fully cite, as this would constitute its own survey paper). In particular, e.g. works on regression for a confidence score for a given answer (https://arxiv.org/abs/2404.15993) or "verbalizing" confidence as the probability mass put on a special token (https://arxiv.org/abs/2412.06676) seem quite related (particular considering my question about differences to regression/classification).

---

> ### Author Rebuttal · Authors · 2025-07-31
>
> Thank you for your kind and constructive feedback on our manuscript! We address your main concerns as follows:
>
> > ### W1 & Q1: Difference between ConfTuner and classification method.
>
> ConfTuner is superior to the classification method for two reasons: first, using LLM to directly verbalize its confidence is **more seamless and intuitive** than having an external classifier generate confidence scores. Second, by fine-tuning the LLM with a tokenized Brier score, **ConfTuner enhances the model's intrinsic calibration ability.** This is evidenced by:
>
> 1) **ConfTuner has good generalization ability to various scenarios and different forms of expressions.** ConfTuner achieves significantly lower ECE scores on out-of-distribution datasets, and also generalizes to different forms of confidence levels (Section 4.2). **We further compare the performance of ConfTuner and the classification method** on the in-distribution and out-of-distribution datasets. We conduct an experiment where we feed the token logits after "Confidence: " from the last hidden state into a two-layer MLP classifier to predict confidence scores from 0 to 100, trained with Brier loss. The results, presented in Tables 1 and 2 below, compare this approach to ConfTuner (based on LLaMA). Our findings indicate that the classifier-based method yields poorer performance compared to ConfTuner, especially on OOD datasets, indicating that **the classifier has relatively higher usage of spurious factors that do not generalize well to diverse scenarios**.
>
> 2) This improved calibration not only boosts trustworthiness but also supports broader LLM capabilities, such as self-correction (Figure 4).
>
> 3) **ConfTuner also generalizes to implicit confidence expressions**, which cannot be done by a classifier. We conduct further tests with a more **open-ended setup**. Specifically, we modified the original prompt from `{Question + Output confidence (0-100)}` to `{Question + Please express your uncertainty when providing the answer}`. This new setting significantly expands the LLM's output space, allowing it to generate diverse natural language expressions of uncertainty, such as "I'm fairly certain, but there’s a chance I could be mistaken" or "This is a tough one, so I’d say it’s likely but not guaranteed." We evaluate these implicit confidences by inputting them to GPT-4o to assess the implied confidence levels. The results of AUROC and ECE are presented in Tables 3 and 4 below, demonstrating that **ConfTuner can handle not only classification-like confidence expressions but also more complex and implicit confidence expressions.
>
> Table 1: Classifier vs ConfTuner comparison (AUROC)
>
> | Method | HotpotQA | GSM8K | TriviaQA | StrategyQA | TruthfulQA | Average |
> |--------|----------|--------|-----------|------------|------------|---------|
> | Classifier | 0.7340 | 0.5843 | 0.6425 | 0.6219 | 0.5648 |  0.6295 |
> | ConfTuner | **0.7383** | **0.7007** | **0.6821** | **0.6750** | **0.5739** | **0.6740** |
>
> Table 2: Classifier vs ConfTuner comparison (ECE)
>
> | Method | HotpotQA | GSM8K | TriviaQA | StrategyQA | TruthfulQA | Average |
> |--------|----------|--------|-----------|------------|------------|---------|
> | Classifier | 0.1253 | 0.2261 | 0.1469 | 0.1526 | 0.3894 |  0.2081 |
> | ConfTuner | **0.0405** | **0.1276** | **0.0388** | **0.1387** | **0.1955** | **0.1082** |
>
> Table 3: Implicit confidence expression performance (AUROC)
> | Method | HotpotQA | GSM8K | TriviaQA | StrategyQA | TruthfulQA | Average |
> |--------|----------|--------|-----------|------------|------------|---------|
> | Base (implicit) | 0.7047 | 0.5422 | 0.6342 | 0.6489 | 0.5895 | 0.6239 |
> | ConfTuner (explicit) | **0.7383** | **0.7007** | 0.6821 | 0.6750 | 0.5739 | 0.6740 |
> | ConfTuner (implicit) | 0.7239 | 0.6869 | **0.7024** | **0.6751** | **0.6217** | **0.6820** |
>
> Table 4: Implicit confidence expression performance (ECE)
>
> | Method | HotpotQA | GSM8K | TriviaQA | StrategyQA | TruthfulQA | Average |
> |--------|----------|--------|-----------|------------|------------|---------|
> | Base (implicit) | 0.2808 | 0.1179 | 0.1232 | **0.1098** | 0.3250 | 0.1913 |
> | ConfTuner (explicit) | **0.0405** | 0.1276 | **0.0388** | 0.1387 | **0.1955** | **0.1082** |
> | ConfTuner (implicit) | 0.1639 | **0.0950** | 0.1088 | 0.1721 | 0.2019 | 0.1483 |
>
> > ### W2: Discussion of two related works.
>
> The regression-based methods [1] rely on an external classifier to calibrate confidence, and can not generalize to different forms or implicit confidence expressions. Similarly, using a single "I don’t know" token limits expressiveness to a binary uncertainty indicator [2], lacking the granularity of our approach. By contrast, ConfTuner fine-tunes the LLM itself, enabling **more accurate and diverse verbalized confidence expressions**. We will include this discussion in our future revision.
>
>
>
> > ### Q2: Would outputting the confidence prediction before the answer change how well the method works?
>
> It’s an interesting experiment and could further elucidate the internal mechanics of ConfTuner. We prompt ConfTuner to generate the confidence score prior to providing the answer. The results of AUROC and ECE are presented in Tables 5 and 6. Our findings indicate that outputting confidence before the answer yields poorer performance compared to outputting it afterward, suggesting that **the hidden states of the answer tokens are indeed informative about the certainty of the response**. And ConfTuner still outperforms Base model when outputting confidence first. We will include this experiment in our revised manuscript.
>
> Table 5: Performance of outputting confidence first (AUROC)
> | Method | HotpotQA | GSM8K | TriviaQA | StrategyQA | TruthfulQA | Average |
> |--------|----------|--------|-----------|------------|------------|---------|
> | Base (confidence+answer) | 0.6909 | 0.5447 | 0.5819 | 0.7094 | 0.4471 | 0.5948 |
> | ConfTuner (confidence+answer) | 0.7263 | 0.6241 | 0.6565 | **0.6787** | 0.5267 | 0.6425 |
> | ConfTuner (answer+confidence) | **0.7383** | **0.7007** | **0.6821** | 0.6750 | **0.5739** | **0.6740** |
>
> Table 6: Performance of outputting confidence first (ECE)
>
> | Method | HotpotQA | GSM8K | TriviaQA | StrategyQA | TruthfulQA | Average |
> |--------|----------|--------|-----------|------------|------------|---------|
> | Base (confidence+answer) | 0.4796 | 0.2082 | 0.1062 | 0.5285 | 0.3761 | 0.3397 |
> | ConfTuner (confidence+answer) | 0.0685 | **0.0953** | 0.1487 | 0.2839 | 0.2889 | 0.1771 |
> | ConfTuner (answer+confidence) | **0.0405** | 0.1276 | **0.0388** | **0.1387** | **0.1955** | **0.1082** |
>
> Thanks again for your valuable suggestions, which are extremely useful for our future work on revising the manuscript.​​ Feel free to reach out if anything comes up. Please let us know if anything needs further clarification!
>
> [1] Uncertainty Estimation and Quantification for LLMs: A Simple Supervised Approach
>
> [2] I Don't Know: Explicit Modeling of Uncertainty with an [IDK] Token

---

> > ### Comment · Reviewer_Fgyi · 2025-07-31
> >
> > Thank you for your response and for providing the additional experiment, the results are indeed interesting!
> >
> > I understand that the format of ConfTuner is more “seamless” than an additional classifier head but I still **do not see the difference between using a classifier and ConfTuner** (except for that in the **classifier experiments** you **do not finetune the rest of the model’s parameters?**). In the end — once we do away with labeling the output embedding rows for the tokens {0,…100} as natural language — these **embeddings of the tokens {0,…100} do literally just form a Dx101 linear layer classifier** (where D is the hidden dim) which is trained via softmax-crossentropy. I.e. if you were to backpropagate gradients from such a classifier into the model, results should be pretty much the same as ConfTuner? The only further difference, which I was alluding to in my review, is that ConfTuner uses the existing pretrained embeddings of {0,…,100} as a good “initialization”.
> > - We could compare how much of a difference this makes (my guess would be that ConfTuner is still a bit better as the initial gradients backpropagated through the classifier will be bad and lead to some catastrophic forgetting — if you use the same initialization for the classifier results should be the same?)
> > - Note that I actually think whichever way these results land would strengthen your work. It might actually be desirable to not “overload” the Arabic numeral embeddings with uncertainty quantification (which might impact e.g. math skills — another potential small but interesting ablation): this could be avoided by using a separate classifier instead of direct fine-tuning of the output embeddings.
> > - Of course, your work introduces the Brier score on top of “run-of-the-mill” classifier for uncertainty quantification, which seems novel in of itself. However, while your method as a whole seems to work well, I do still think there is more we can do to actually probe the specific mechanisms behind it’s good performance to really push it to the next level — e.g. with some of the experiments I mention above.
> >
> > One more comment: the IDK-token paper actually also does make some remarks about generalization to verbalized quantifications of uncertainty even though training only tuned their single IDK-token (in Section 4.5 of that paper, although the evaluation is less robust). I think in principle, this might be expected of many methods which finetune an LLM to quantify uncertainty in its responses. One hypothesized mechanism could be a promotion of already existing “provide uncertainty caveat” directions in latent space (similar to a “refusal direction” [1]). This does not detract from your method. Based on your additional experiments provided in response to multiple Reviewers, there is reason to believe that the format you choose does most effectively “trigger” this generalization mechanism, potentially because it is much closer to natural language.
> >
> > While I am **not raising my score as of now**, I do think this **work has merit and is interesting**. Furthermore, I will **raise my score** if my points regarding the **(lack of) difference between ConfTuner and a linear classification head are sufficiently addressed**.
> >
> > [1] https://arxiv.org/abs/2406.11717

---

> > > ### Author Response · Authors · 2025-08-03
> > > **Response to Reviewer Fgyi**
> > >
> > > Thanks for your insightful suggestion! We address your concers as follows:
> > >
> > >
> > > ### Difference between ConfTuner and classifier
> > >
> > > In the following, our goal is to do a precise comparison between (A) ConfTuner, and (B) an LLM with an external linear classifier with the same architecture as the model's original output projection layer. Specifically, this confidence classifier is a linear transformation layer, whose input dimension matches the dimension of the model's hidden states, and its output dimension equals the size of the model's vocabulary. The input to this classifier is the final hidden state from the LLM's last layer, corresponding to the last token position in the generated sequence.
> > >
> > >
> > > We would like to clarify that for our confidence estimation setting, both (A) and (B) have to be trained using our Brier loss, rather than the standard cross-entropy loss, since there are no ground truth confidence labels available for the confidence estimation task.
> > >
> > >
> > > Given the above, we agree with the reviewer's understanding that (A) and (B) have the exact same architecture, and the only differences between them are (1) **End-to-end training**: in (A), we train the LLM end to end, but in (B) we train only the final linear laye. (2) **Initialization / parameter sharing**: in (A), the output projection layer parameters are tied with the LLM's original embedding matrix, while in (B), the classifier's parameters are not tied and randomly initialized.
> > >
> > > To further investigate the impact of two differences, we conduct experiments on (A) ConfTuner, (B) the classifier with random initialization, and (C) the classifier initialized with the LLM's original embedding matrix. We found that **end-to-end training and initialization are both important for the calibration performance**. Specifically, as shown in Table 1 and Table 2, we have the following observations: (1) the classifier initialized with LLM’s original embedding matrix performs better than the classifier with random initialization. This indicates that the random initialization might lead to noise (or noisy gradients), resulting in sub-optimal results. (2) ConfTuner still performs better than the classifier initialized with LLM’s original embedding matrix. This is because the classifier infers only based on the hidden state of the LLM. If the final hidden state does not capture sufficient information about the model's confidence, the classifier will be less effective at confidence estimation. In contrast, ConfTuner trains the LLM itself's parameters, so the LLM can be trained to preserve the necessary confidence information in the final hidden state.
> > >
> > >
> > > Table 1: AUROC
> > > | Method | HotpotQA | GSM8K | TriviaQA | StrategyQA | TruthfulQA | Average |
> > > |-|-|-|-|-|-|-|
> > > | classifier+random initialization | 0.6817 | 0.6025 | 0.6442 | 0.5961 | 0.5428 | 0.6335 |
> > > | classifier+llm initialization | 0.7356 | 0.6518 | **0.6873** | 0.6420 | 0.5626 | 0.6559 |
> > > | ConfTuner | **0.7383** | **0.7007** | 0.6821 | **0.6750** | **0.5739** | **0.6740** |
> > >
> > > Table 2: ECE
> > > | Method | HotpotQA | GSM8K | TriviaQA | StrategyQA | TruthfulQA | Average |
> > > |-|-|-|-|-|-|-|
> > > | classifier+random initialization | 0.0865 | 0.2983 | 0.1582 | 0.2057 | 0.2493 | 0.1996 |
> > > | classifier+llm initialization | 0.0581 | 0.1685 | 0.0621 | 0.1459 | 0.2206 | 0.1310 |
> > > | ConfTuner | **0.0405** | **0.1276** | **0.0388** | **0.1387** | **0.1955** | **0.1082** |
> > >
> > >
> > > ### Impact on math problems
> > >
> > > To directly evaluate this concern, ​​we compared the mathematical reasoning performance of base LLMs versus ConfTuner on GSM8K​​, a challenging math word problem dataset. The results are summarized in Table 3.  Across models, accuracy changes are ​​minimal (±≤1.5%)​​ - well within typical fine-tuning variance. ConfTuner maintains competitive performance, with ​​Ministral even showing slight improvement (+1.43%). In the future, we will explore whether freezing the projection layer during ConfTuner fine-tuning​​ could alleviate this impact.
> > >
> > > Table 3: Accuracy
> > > | LLM | Base | ConfTuner | change rate |
> > > |-|-|-|-|
> > > | LLaMA | 0.7970 | 0.7850 |-1.51% |
> > > | Qwen | 0.8680 | 0.8620 | -0.69% |
> > > | Ministral | 0.6980 | 0.7080 | +1.43% |
> > >
> > > ### Discussion about the IDK-token paper
> > >
> > > Thank you for this insightful remark​​! We fully agree with your observation that methods like ConfTuner—and indeed other fine-tuning approaches—may ​​amplify pre-existing latent directions for uncertainty expression​​, akin to the "refusal direction" phenomenon discussed in prior work. And we agree with your hypothesis that **our verbalized uncertainty format effectively ​​activates these generalized uncertainty pathways​​ because its natural language grounding is particularly valuable**. Linguistic alignment may play a key role in triggering robust uncertainty generalization. We will integrate your commentary​​ into our future revision.
> > >
> > > We'd be delighted to engage with any follow-up questions or discussions this may inspire.

---

> > > > ### Comment · Reviewer_Fgyi · 2025-08-03
> > > >
> > > > Thank you for your engagement in this discussion.
> > > >
> > > > Your response makes very clear the distinctions and similarities between your method and training a linear classifier and you back up the differences with experimental results.
> > > > Your experiments re: Math ability are a valuable answer to a potential drawback.
> > > > Additionally, your points re: latent uncertainty pathways are interesting and — in my opinion — also point the way for interesting follow-up work.
> > > >
> > > > I think it is valuable to feature this discussion and results prominently in the revised draft, as it will help give readers a good understanding of the important ingredients to your method rather than a — spoken candidly, at times vague — reference to „natural language verbalisation“.
> > > >
> > > > Overall, I commend the authors for their stellar engagement in the review process. In my opinion, their work has been meaningfully improved as a result. Therefore, I recommend the work to be accepted, raise my score to 5 — `Accept`, and hope the authors integrate the substantial new experiments and discussions of this reviewing phase into their paper before publication.

---

> ### Author Response · Authors · 2025-08-03
> **Response to Reviewer Fgyi**
>
> Thank you sincerely for your ​​thoughtful and constructive feedback​​ throughout this process! Your insights and suggestions will ​​significantly enhance our work​​. All new experiments and discussions during this review cycle will be ​​rigorously integrated​​ into the final paper. Specifically, we will:
>
> ​​1. Expand Section 4.5​​ to discuss ConfTuner's underlying mechanisms (comparison to classifier), emphasizing its ​​system-level confidence calibration capabilities​​ beyond verbalized uncertainty expressions.
>
>
> 2. ​​Incorporate new analysis​​ on mathematical robustness (GSM8K results in Table 3) to reinforce our reliability claims.
>
> We are profoundly grateful for your decision to ​​raise the score to 5 (Accept)​​. Your engagement has been ​​exceptionally valuable​​ in advancing this research.

---

### Official Review · Reviewer_gPzL · 2025-07-02

**Clarity:** 3
**Significance:** 2
**Originality:** 3
**Rating:** 5
**Confidence:** 2

**Summary:**

The paper presents a loss function for training LLMs to be better calibrated in their confidence assessments. The paper presents a theoretical argument for the proposed loss function being a Proper Scoring Rule and empirical testing on a combination of benchmark datasets with different tasks, different models, and different verbalizations of confidence. The paper finds that the ConfTuner loss consistently provides better confidence estimates, including out of domain, relative to other techniques or just using a base model’s estimates.

**Questions:**

I do have a few questions about the paper

1.	For the LoRA training, what was the alpha, and what layers of the neural networks did you attach the adapters to?
2.	Could the proposed loss be combined with other losses to improve both the quality of the answers and the calibration in the same training?
3.	Does the confidence calibration carry over to tasks/prompts where there aren’t explicit instructions to evaluate its confidence? In other words, is there any implicit confidence calibration that is learned, such that a prompt that includes things like “…If you don’t know something or are unsure about something, do …” would work better? I think it's just a little clunky to always have the LLM output its confidence on everything, and it would be nice if it had better implicit confidence calibration.

**Ethical Concerns:**

["NO or VERY MINOR ethics concerns only"]

**Final Justification:**

I will stand by my rating of 'Accept' after participating in the discussion period and reviewing the other comments. I appreciate that the authors' addressed some of the clarity issues with their testing setup and believe the paper does make a contribution.

**Limitations:**

All limitations are successfully addressed in the paper.

**Quality:**

4

**Strengths And Weaknesses:**

The paper has a strong empirical validation and is well written and easy to understand. I especially appreciate that the paper did out-of-domain testing (i.e., they trained confidence assessments for one dataset and task and then showed how that training carried over to new datasets and tasks). I believe that such a result shows promise for practical usage. The proposed method is also simple to implement, with good theoretical guarantees.

The weaknesses of the paper are few. I suppose one could argue that the problem of getting accurate confidence estimates from LLMs is of secondary importance to improving the general, and especially reasoning, performance of LLMs. This paper is not likely to be ground-breaking for the field, but I still think it shows a lot of promise as a technique for actual usage.

---

> ### Author Rebuttal · Authors · 2025-07-31
>
> Thank you for your encouraging feedback and insightful comments! We address your main concerns as follows:
>
> > ### W1: Importance of confidence estimation vs core LLM advancement​
>
> Improving the general capabilities of LLMs is indeed important. However, **even as LLMs become more capable, there will inevitably be questions that exceed their knowledge or reasoning abilities**. In such cases, the model’s tendency to be overconfident or fail to accurately express their uncertainty is a fundamental limitation, undermining user trust and making it hard for downstream systems or users to know when caution is warranted.
>
> Moreover, **the ability to express calibrated confidence can also support better reasoning and decision-making**. For instance, in retrieval-augmented generation (RAG) or tool-use settings, a model that can identify and verbalize its uncertainty within its chain-of-thought could trigger additional information-seeking steps, such as searching for more evidence or invoking a tool, to validate or revise its initial answer, allowing for iterative self-correction and better performance.
>
>
>
> > ### Q1: LoRA setting.
>
> The alpha value is set to 32, with adapters applied to all layers - specifically attached to the query and value projection modules. We will incorporate this implementation detail in our manuscript revision.
>
> > ### Q2: Could the proposed loss be combined with other losses to improve both the quality of the answers and the calibration in the same training?
>
> Yes. As a plug-and-play loss, the Brier score is highly flexible and compatible with standard LLM fine-tuning pipelines. **Brier loss can be seamlessly combined with other loss functions, such as those optimizing answer quality**, to enhance both calibration and response accuracy during training simultaneously. In this work, we focused exclusively on improving verbalized confidence calibration, so we only use the Brier loss. Future work could explore integrating it with answer-focused losses to achieve dual improvements.
>
> > ### Q3: Does the confidence calibration carry over to tasks/prompts where there aren’t explicit instructions to evaluate its confidence?
>
> **ConfTuner could generalize to scenarios without explicit instructions.** We conduct new experiments on ConfTuner (based on LLaMA). Specifically, we change the original prompt from `{Question + output confidence (0-100)}` to `{Question + Please express your uncertainty when providing the answer}`. Under this instruction, the LLM also produces implicit confidence expressions, such as "I'm fairly certain, but there’s a chance I could be mistaken" or "This is a tough one, so I’d say it’s likely but not guaranteed." We evaluate these implicit confidence by inputting them to GPT-4o to assess the implied confidence levels (0-100). The results of AUROC and ECE are shown in Table 1 and Table 2, demonstrating that **implicit confidence calibration of ConfTuner is comparable to explicit confidence calibration.**
>
> Table 1: AUROC
> | Method | HotpotQA | GSM8K | TriviaQA | StrategyQA | TruthfulQA | Average |
> |--------|----------|--------|-----------|------------|------------|---------|
> | Base (implicit) | 0.7047 | 0.5422 | 0.6342 | 0.6489 | 0.5895 | 0.6239 |
> | ConfTuner (explicit) | **0.7383** | **0.7007** | 0.6821 | 0.6750 | 0.5739 | 0.6740 |
> | ConfTuner (implicit) | 0.7239 | 0.6869 | **0.7024** | **0.6751** | **0.6217** | **0.6820** |
>
> Table 2: ECE
>
> | Method | HotpotQA | GSM8K | TriviaQA | StrategyQA | TruthfulQA | Average |
> |--------|----------|--------|-----------|------------|------------|---------|
> | Base (implicit) | 0.2808 | 0.1179 | 0.1232 | **0.1098** | 0.3250 | 0.1913 |
> | ConfTuner (explicit) | **0.0405** | 0.1276 | **0.0388** | 0.1387 | **0.1955** | **0.1082** |
> | ConfTuner (implicit) | 0.1639 | **0.0950** | 0.1088 | 0.1721 | 0.2019 | 0.1483 |
>
> Thank you again for your supportive review. Your constructive comments are greatly appreciated for revising our paper. We welcome any additional questions you might have.

---

> > ### Comment · Reviewer_gPzL · 2025-08-05
> > **Response to Rebuttal**
> >
> > I appreciate the authors adding some extra details, as well as the implicit confidence experiment. In mind, this strengthens the utility of the proposed training. After having read through the other reviews and responses, I still believe this paper is an "Accept" and will maintain that rating.

---

> > > ### Author Response · Authors · 2025-08-06
> > > **Response to Reviewer gPzL**
> > >
> > > Thank you very much for your constructive suggestions and continued support for our manuscript. We greatly appreciate your positive comments regarding the additional details and the implicit confidence experiment we added.
> > > We plan to incorporate the discussion and new experiments in our ​​future work​​ to further enhance the paper. Your feedback is indeed crucial for the improvement and overall ​​enhancement​​ of our research.

---

### Decision · Program_Chairs · 2025-09-17

**Decision:**

Accept (poster)

**Comment:**

This paper aims to address the overconfidence problem of LLMs and focuses on calibrating LLM's verbalized confidence. It proposes that existing approaches, which rely on prompt engineering or fine-tuning with heuristical uncertainty estimations, have limited effectiveness and generalizability. To mitigate this issue, it proposes a new loss function tokenized Brier score and a method ConfTuner. The experimental results demonstrate that the proposed method improves calibration across multiple tasks and foundation models. The reviewers comment that this paper is well-motivated, has certain theoretical and empirical contributions, and is easy to follow. The authors have adequately address the proposed concerns, such as highlighting the difference between existing methods, adding more discussions and explanations, and including necessary baseline methods. After the rebuttal, all of the reviewers give positive comments and ratings. Therefore, I recommend accepting this paper.